# Modeling Transitivity and Cyclicity in Directed Graphs via Binary Code Box Embeddings

**Dongxu Zhang, Michael Boratko, Cameron Musco, Andrew McCallum**
University of Massachusetts Amherst
{dongxuzhang, mboratko, cmusco, mccallum}@cs.umass.edu

## Abstract

Modeling directed graphs with differentiable representations is a fundamental requirement for performing machine learning on graph-structured data. Geometric embedding models (e.g. hyperbolic, cone, and box embeddings) excel at this task, exhibiting useful inductive biases for directed graphs. However, modeling directed graphs that both contain cycles and some element of transitivity, two properties common in real-world settings, is challenging. Box embeddings, which can be thought of as representing the graph as an intersection over some learned super-graphs, have a natural inductive bias toward modeling transitivity, but (as we prove) cannot model cycles. To this end, we propose *binary code box embeddings*, where a learned binary code selects a subset of graphs for intersection. We explore several variants, including global binary codes (amounting to a union over intersections) and per-vertex binary codes (allowing greater flexibility) as well as methods of regularization. Theoretical and empirical results show that the proposed models not only preserve a useful inductive bias of transitivity but also have sufficient representational capacity to model arbitrary graphs, including graphs with cycles.

## 1 Introduction

Many real-world networks, such as social media interactions, paper citations, web links, and ontologies, are naturally represented as directed graphs [14, 6]. Two common properties of these graphs are transitivity and cyclicity. A *cycle* in a directed graph is a directed path starting from a vertex and traversing back to itself. For example, "organic matter" $\rightarrow$ "worm" $\rightarrow$ "fish" $\rightarrow$ "cat" $\rightarrow$ "organic matter" is a food cycle. *Transitivity* in a directed graph is the property that if there exists a directed path from $u$ to $v$, then edge $(u, v)$ also exists. For example, if "cat is mammal" and "mammal is animal" are true, "cat is animal" is true.

In the age of deep learning, it is necessary to determine a way to capture the salient information from a graph via some differentiable parameterization. To this end, various graph embedding methods have been proposed. Some work, such as DeepWalk [17] and Node2vec [8], maps each vertex to a vector in Euclidean space. These methods perform a low-rank factorization of the adjacency matrix or the graph Laplacian [18] and are designed to model *undirected* graphs. These can be extended to capture edge asymmetry in *directed* graphs by using two separate representations - one for source, and one for target - either unconstrained or related with one another via some function. It has been proven that using a dot product or distance-based energy function, separate source and target vectors can encode any graph (given sufficient dimension) [1], including graphs with cycles. When the source and target representations live in separate spaces, however, relationships which result from following directed paths (e.g., transitivity) are harder to encode. For example, given edge $(i, j)$ and $(j, k)$, a vector model can learn to make $|s_i - t_j|_2 \approx 0$ and $|s_j - t_k|_2 \approx 0$, where $s_i, s_j$ are source vectors and

36th Conference on Neural Information Processing Systems (NeurIPS 2022).

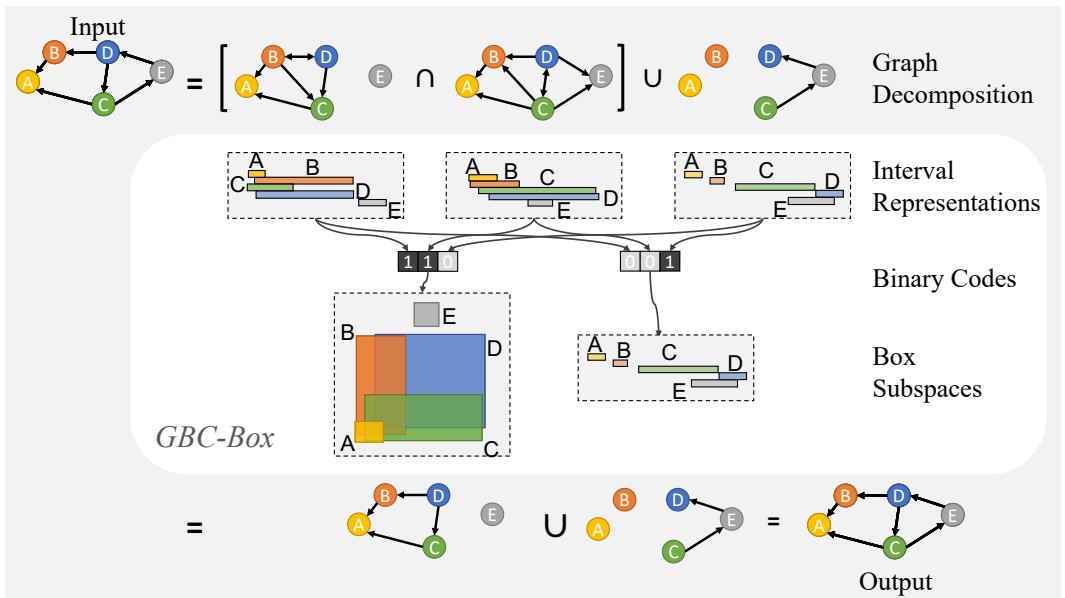

Figure 1: Demonstration of using GBC-BOX to represent a directed graph. Firstly, the model learns $d(=3)$ interval embeddings, each representing a directed graph. Then, there are $k(=2)$ binary code vectors, each of which selects a subset of the interval embeddings to combine into a box embedding (in this case, 2-D and 1-D) subspace. In each subspace, box embeddings can represent a DAG as shown in the bottom of the diagram. Finally, the target graph can be reconstructed as a union over DAGs from all subspaces.

$t_j, t_k$ are target vectors. However, the condition $|s_i - t_k|_2 \approx 0$ which would represent the transitive edge $(i, k)$ has no encouragement to hold, as the source and target spaces are entirely disconnected.

Transitivity cannot be trivially injected via symbolic rules, e.g., adding all transitive edges to a directed graph. This is because most real world edges are not *strictly* transitive: the degree of transitivity is "soft", and may hold locally but not globally, or vary for different edge types or sets of nodes in the graph. Some work, such as HOPE [15] and APP [29], attempts to capture transitivity by factorizing high-order proximity signals instead of the graph Laplacian. This branch of work is limited by the imperfection of high-order proximity scores (for example, these scores do not model cycles well). ATP [25] resolves this by breaking cycles in the graph, accepting a loss of graph information.

An alternative approach is to represent nodes in an embedding space with additional geometric structure. For example, hyperbolic embeddings leverage the negative curvature of hyperbolic space to provably model trees with less distortion [22]. Region-based embeddings such as box embeddings [28, 11, 3] and hyperbolic entailment cones [13, 7, 21, 9] have a natural bias toward modeling transitivity. These region-based embeddings can capture the transitivity in a directed graph without relying on high-order proximity scores, using only the original adjacency matrix as supervision. Previous work [2] proved that box embeddings can model any directed acyclic graph (DAG). A natural question, therefore, is whether box embeddings can capture graphs with cycles. As we will show in Section 2, this is not the case.

In this work, we propose *binary code box embeddings* [1], a generalization of box embeddings to represent arbitrary directed graphs. The model is motivated by the intuition that a given directed graph can be regarded as a union of multiple sub-graphs, where each sub-graph is acyclic, and therefore can be represented using boxes. We introduce the concept of a "binary code" which selects these sub-graphs.

Our contributions lie in three folds:

- We propose global (GBC-BOX) and per-vertex (VBC-BOX) binary code box models, a generalization of box embeddings capable of representing arbitrary directed graphs.

---

[1]Our code is available at https://github.com/iesl/geometric_graph_embedding

- We analyze theoretically the limitations of existing box embedding models when representing cycles. We also prove that, given sufficient dimensions, both binary code box models can model any directed graph. This establishes that, in theory, the representational capacity of these models is not limited.

- We evaluate our model on graph reconstruction and link prediction tasks with various synthetic graphs and real world graphs, and observe that our proposed methods perform the best in almost all scenarios, especially when a graph has strong cyclicity and transitivity.

## 2  Background

Given a simple[2] directed graph $G$ with vertices and edges $(\mathcal{V}, \mathcal{E})$, we seek to represent the vertices using some mapping $\phi : \mathcal{V} \to \mathbb{R}^d$, and an energy function $\mathrm{E} : \mathcal{V} \times \mathcal{V} \to \mathbb{R}_+$ providing a score (based on $\phi$ and, perhaps, some hyperparameters $\boldsymbol{\lambda}$) which is interpreted as the negative log probability of edge existence, $\mathrm{E}(u, v; \phi, \boldsymbol{\lambda}) = -\log P((u, v) \in \mathcal{E})$. We can view these probabilities as a weighted graph, however in practice it is often necessary to make a hard decision on edge existence, which is done by choosing a (global) threshold $T$ and binarizing the output. We denote the energy function for a particular model M with a subscript, i.e. $\mathrm{E_M}$, and say that M is capable of modeling a graph $G = (\mathcal{V}, \mathcal{E})$ if there is some $\phi$, $\boldsymbol{\lambda}$, and $T$ such that $\mathcal{E} = \{(u, v) \mid \mathrm{E_M}(u, v; \phi, \boldsymbol{\lambda}) < T\}$.

### 2.1  Boxicity

Let $\mathbf{I}$ be the set of closed and bounded intervals in $\mathbb{R}$. An *interval graph* is an undirected graph $G = (\mathcal{V}, \mathcal{E})$ such that there exists a mapping $\varphi : \mathcal{V} \to \mathbf{I}$ for which

$$\{u, v\} \in \mathcal{E} \quad \Longleftrightarrow \quad \varphi(u) \cap \varphi(v) \neq \emptyset.$$

More generally, we can consider $\mathbf{B}^d$, the set of $d$-dimensional "boxes", which are Cartesian products of intervals,

$$\prod_{i=1}^d \left[x_i^-, x_i^+\right] = \left[x_1^-, x_1^+\right] \times \cdots \times \left[x_d^-, x_d^+\right] \subseteq \mathbb{R}^d. \tag{1}$$

where $x_i^-$ and $x_i^+$ are min and max coordinates in dimension $i$. As defined by Roberts [20], the *boxicity* of an undirected graph $G$ is the smallest dimension $d$ such that there exists a mapping $\varphi : \mathcal{V} \to \mathbf{B}^d$ for which $\{u, v\} \in \mathcal{E}$ if and only if $\varphi(u) \cap \varphi(v) \neq \emptyset$. Equivalently, the boxicity is the minimal number of interval graphs whose intersection is $G$.

### 2.2  Box Embeddings

Vilnis et al. [28] provide a way of using boxes to represent directed graphs by defining an asymmetric energy function based on box volumes, and subsequent work has introduced various improvements and extensions of this idea Dasgupta et al. [4], Boratko et al. [2]. In this section we will define the energy function of box embedding models under a common framework, in preparation to motivate our extension to binary code box embeddings.

The energy function for all box embedding variants has the form

$$\mathrm{E}(u, v; \phi, \boldsymbol{\lambda}) = -\log \prod_{i=1}^d F(\phi(u)_i, \phi(v)_i; \boldsymbol{\lambda}), \tag{2}$$

where $\phi(u)_i$ are the parameters associated with node $u$ in dimension $i$ and $F(\phi(u)_i, \phi(v)_i; \boldsymbol{\lambda}) \in [0, 1]$ is a per-dimension score representing the probability of edge existence. [3] The model originally defined in Vilnis et al. [28] represented each node using a box, as in (1). The per-dimension parameters are the endpoints of an interval, $\phi(u)_i = [\phi(u)_i^-, \phi(u)_i^+]$, and the score function is defined as

$$F_{\mathrm{Box}}([x^-, x^+], [y^-, y^+]) = \frac{\lvert [x^-, x^+] \cap [y^-, y^+] \rvert}{\lvert [y^-, y^+] \rvert} = \frac{\max(\min(x^+, y^+) - \max(x^-, y^-), 0)}{\max(y^+ - y^-, 0)}.$$

---

[2] A *simple* directed graph is one without multiple edges or self-loops, i.e. the adjacency matrix contains only 0s and 1s, with 0s on the diagonal. We also synthetically remove self-loops from the graph modeled by the learned representations.

[3] As in boxicity, we can interpret these box embedding models as representing a graph as an intersection of interval graphs, one for each dimension.

This encourages the box for a given vertex to contain (or overlap highly with) the boxes for it's children. Boxes which are disjoint or contained in one another can present problems for training, however. Dasgupta et al. [3] addressed this by modeling the endpoints of intervals using Gumbel random variables. The per-dimension score can be written as[4]

$$F_{\text{G-Box}}((x^-, x^+), (y^-, y^+); (\tau, \nu)) = \frac{\text{LSE}_\nu\left(-\text{LSE}_\tau(-x^+, -y^+) - \text{LSE}_\tau(x^-, y^-), 0\right)}{\text{LSE}_\nu(y^+ - y^-, 0)}, \quad (3)$$

where $\text{LSE}_t$ denotes the following continuous extension of LogSumExp with temperature $t \geq 0$:

$$\text{LSE}_t(\mathbf{x}) = \begin{cases} t\log(\sum_i e^{x_i/t}) & \text{if} \quad t > 0, \\ \max(\mathbf{x}) & \text{if} \quad t = 0. \end{cases} \quad (4)$$

In practice, the temperatures $\tau$ and $\nu$ are tuned separately as global hyperparameters, however when they are equal the parameters $x^-, y^-$ (resp. $x^+, y^+$) can be interpreted as the mean of the GumbelMax (resp. GumbelMin) random variables with scale $\nu = \tau$, and $F_{\text{G-Box}}$ approximates a ratio of expected box volumes. Note that for any $(x^-, x^+), (y^-, y^+) \in \mathbb{R}^2$, $F_{\text{G-Box}}$ is continuous with respect to $\tau, \nu \in \mathbb{R}_{\geq 0}$, and $F_{\text{G-Box}}((x^-, x^+), (y^-, y^+); 0, 0) = F_{\text{Box}}([x^-, x^+], [y^-, y^+])$.

Boratko et al. [2] takes this one step further, using a mapping which learns 4 parameters per dimension: $\phi(u)_i = (\phi(u)_i^-, \phi(u)_i^+, \phi(u)_{i,\tau}, \phi(u)_{i,\nu})$. The per-dimension score function is then defined as

$$F_{\text{T-Box}}((x^-, x^+, x_\tau, x_\nu), (y^-, y^+, y_\tau, y_\nu)) = F_{\text{G-Box}}((x^-, x^+), (y^-, y^+); (\tfrac{x_\tau + y_\tau}{2}, \tfrac{x_\nu + y_\nu}{2})). \quad (5)$$

## 3 Existing Representational Capacity and Limitations

### 3.1 Representational Capacity

We would like to know each model's representational capacity (the set of graphs capable of being modeled) as well as how this may change depending on hyperparameter settings. It was proven in Boratko et al. [2] that BOX can model any DAG, and of course since T-BOX is equivalent to BOX when $\phi(u)_{i,\tau} = \phi(u)_{i,\nu} = 0$ this also holds for T-BOX. As defined in Section 2, we can also say that G-BOX is capable of modeling any DAG, since it is equivalent to BOX when we set the temperature hyperparameters to zero (i.e. $\tau = \nu = 0$), however this is practically quite different – the temperatures are not trainable as in T-BOX, and would never be set to 0 in order to avoid the training difficulties of BOX. Hence, the existing proof of representational capacity in [2] says very little about the practical representational capacity of G-BOX.[5] Thankfully, more can be proven.

**Theorem 1.** *Given a threshold $T$, temperature hyperparameters $\tau, \nu$, there exists and a bijection $\psi$ on the set of parameterizations $\{\mathcal{V} \to \mathbb{R}^{2d}\}$ such that for all $u, v \in \mathcal{V}$,*

$$\text{E}_{\text{G-Box}}(u, v; \psi(\phi), (\tau, \nu)) < T \quad \Longleftrightarrow \quad \text{E}_{\text{Box}}(u, v; \phi) < T.$$

In other words, for *any* temperature hyperparameters, G-BOX can represent any graph representable by BOX. For the proof, see Appendix B. Proposition 3 in Boratko et al. [2] states that any DAG can be modeled by BOX in $\mathcal{O}((\Delta + 2)\log|\mathcal{V}|)$ dimensions with $\mathcal{O}(D(\Delta + 2)\log^2|\mathcal{V}|)$ bits of precision per box, where $D \leq |\mathcal{V}|$ is the depth of $G$ and $\Delta$ is the maximum degree. Combining this with Theorem 1 we have the following:

**Corollary 2.** *Let $G$ be any DAG. Given any $\tau, \nu \in \mathbb{R}_{\geq 0}$, there exists a mapping $\phi : \mathcal{V} \to \mathbb{R}^{2d}$ and a threshold $T > 0$ such that $\text{E}_{\text{G-Box}}(u, v; \phi, (\tau, \nu)) < T$ if and only if $(u, v) \in \mathcal{E}$, where $d = \mathcal{O}((\Delta + 2)\log|\mathcal{V}|)$, and $\Delta$ is the maximum degree in $G$.*

In other words, for any setting of temperature hyperparamters, G-BOX can model any DAG.

---

[4]In Dasgupta et al. [4] they interpret $\phi(u)_i^\pm$ as the "location" parameters of the distribution, resulting in a slightly different form of the score function, however as we show in Appendix A our score (3) leads to an equivalent model.

[5]This limitation was acknowledged in Boratko et al. [2] just before Section 4.3 as part of the motivation for the trainable temperatures of T-BOX.

## 3.2 Limitation on Modeling Cycles

In this section, we point out that G-Box cannot model any graph containing a (directed) *chordless cycle*, which is a cycle such that no two vertices are connected by an edge which does not belong to the cycle.

**Theorem 3.** *If* E *is such that* $\mathrm{E}(u,v) - \mathrm{E}(v,u) = g(u) - g(v)$ *for some function* $g : \mathcal{V} \to \mathbb{R}$ *then it cannot model any graph containing a chordless cycle with more than 2 nodes.*

*Proof.* Suppose the vertices $1, 2, ..., N$ comprise a chordless cycle, the edges of which are $D = \{(1, 2), (2, 3), ..., (N - 1, N), (N, 1)\}$. Suppose further that we can model the graph containing this cycle using energy E and threshold $T$. In particular, we have $\mathrm{E}(u, v) < T$ and $\mathrm{E}(v, u) \geq T$ for $(u, v) \in D$. This implies that $g(u) - g(v) = \mathrm{E}(u, v) - \mathrm{E}(v, u) < 0$, and thus $g(u) < g(v)$ for each $(u, v) \in D$. Hence $g(1) < g(2) < \cdots < g(N) < g(1)$, which is a contradiction. $\square$

**Corollary 4.** G-Box *cannot model any graph containing a chordless cycle on more than 2 nodes.*

*Proof.* Theorem 3 applies to $E_{\text{G-Box}}$ with $g(u) = -\log \prod_{i=1}^{d} \mathrm{LSE}_{\nu}(\phi(u)_i^+ - \phi(u)_i^-, 0)$. $\square$

It is possible to avoid the contradiction in Theorem 3 by the introduction of one reverse edge, and in this case we observe that it is theoretically possible for G-Box to represent a graph.

**Proposition 1.** *If* $G$ *is a graph which is the union of a chordless cycle and one reverse edge,* G-Box *can model* $G$ *in 2 dimensions.*

The proof of this statement is contained in Appendix C.

## 4 Method

In this section, we will introduce the binary code box embedding concept, which includes a family of models whose shared feature is the use of learned binary codes to select subsets of dimensions. Two we will focus on in particular include GBC-Box, which uses "global" binary code vectors, and VBC-Box, which uses per-vertex binary code vectors. The following topics will be covered: the motivation for binary codes, the definition of GBC-Box and VBC-Box, their representational capacity to model arbitrary directed graphs, our learning objectives and regularization, the models' inductive biases and strengths, and some discussion about their limitations and alternative variants.

### 4.1 Motivation

The idea of binary code boxes is to allow more flexibility than simply taking an intersection over interval graphs, as captured by boxicity (see Section 2.1). Recall, in the undirected case, the boxicity of a graph $G = (\mathcal{V}, \mathcal{E})$ was equivalent to the smallest number $d$ such that for some set of interval graphs $S = \{G_i = (\mathcal{V}, \mathcal{E}_i)\}_{i=1}^{d}$, we have $\mathcal{E} = \cap_{i=1}^{d} \mathcal{E}_i$, i.e.

$$\{u, v\} \in \mathcal{E} \iff \forall \mathcal{F} : (\mathcal{V}, \mathcal{F}) \in S, \quad \{u, v\} \in \mathcal{F}.$$

There are various ways to increase the flexibility of this representation. For example, we could consider allowing a union over intersections by specifying $k$ subsets of $S$, $\{S_i\}_{i=1}^{k}$, for which $\mathcal{E} = \cup_{i=1}^{k} \cap_{(\mathcal{V}, \mathcal{F}) \in S_k} \mathcal{F}$, i.e.

$$\{u, v\} \in \mathcal{E} \iff \exists i : \forall \mathcal{F} : (\mathcal{V}, \mathcal{F}) \in S_i, \quad \{u, v\} \in \mathcal{F}. \tag{6}$$

To allow for even greater flexibility, we could allow each vertex to select a subset of graphs to intersect. Formally, we allow the specification of a function $\psi : \mathcal{V} \to 2^S$ which assigns a subset of interval graphs to each vertex, for which

$$\{u, v\} \in \mathcal{E} \iff \forall \mathcal{F} : (\mathcal{V}, \mathcal{F}) \in \psi(u) \cap \psi(v), \quad \{u, v\} \in \mathcal{F}. \tag{7}$$

For the undirected case the advantage is minimal. Increasing the flexibility in these ways may allow us to represent an undirected graph in smaller dimension (or, equivalently, using a smaller number of interval graphs), however as mentioned previously we know any undirected graph can be represented as an intersection of interval graphs. For directed graphs, however, this is not the case, and (as we prove in Section 4.3) this generalization allows for any directed graph to be represented.

## 4.2 Definition

In order to capture the idea of a "union of intersections" specified in (6) we consider learning $k$ "binary code" vectors $\mathbf{b}_j \in [0,1]^d$. Each binary vector corresponds to a selection of which dimensions to include - if the $i^{\text{th}}$ component is 0 the scores for edges in this dimension should be ignored, and if it is 1 they should be included. For convenience, we will represent these as the columns of a $d \times k$ matrix $B \in [0,1]^{d \times k}$. The energy function in this case is

$$\text{E}_{\text{GBC-Box}}(u,v;(\phi,B),(\tau,\nu,k)) := \min_{j \in \{1,\ldots,k\}} \left( -\log \prod_{i=1}^d F_{\text{G-Box}}(\phi(u)_i, \phi(v)_i; (\tau,\nu))^{B_{i,j}} \right) \quad (8)$$

In order to capture the notion of per-vertex subset selection in (7), we learn 3 parameters per dimension, which we denote as $\phi(u)_i = (\phi(u)_i^-, \phi(u)_i^+, \phi(u)_i^\diamond) \in \mathbb{R} \times \mathbb{R} \times [0,1]$. The binary code $\phi(u)_i^\diamond$ indicates whether this dimension should be taken into account when calculating the probability of edge existence for edges involving this node - if it is 0, the dimension should be ignored, and if it is 1 it may be included. We incorporate this at the level of the per-dimension score function as follows:

$$F_{\text{VBC-Box}}((x^-, x^+, x^\diamond), (y^-, y^+, y^\diamond); (\tau,\nu)) := F_{\text{G-Box}}((x^-, x^+), (y^-, y^+); (\tau,\nu))^{x^\diamond y^\diamond}. \quad (9)$$

Using the product in the exponent is a relaxation of the intersection $\psi(u) \cap \psi(v)$ from (7). When computing the probability of an edge $(u,v)$, the binary codes can learn to ignore certain dimensions by making $\phi(u)_i^\diamond$ or $\phi(v)_i^\diamond$ equal to 0.

In the following, we point out two perspectives which provide further intuition behind these models:

**Generalization:** Both GBC-Box and VBC-Box are a generalization of G-Box, as the energy function is equivalent when all binary codes are 1, in which case all dimensions are used for volume calculation. As we will show in Section 4.3 these models are strictly more expressive, as when some binary codes are less than 1 these models can represent more complex graphs.

**Projection:** For GBC-Box, if $B_{i,j} \in \{0,1\}$ we can think of this defining a set of projections $\{P_j\}_{j=1}^k$ where $P_j$ projects the boxes parameterized by $\phi$ into the $\sum_{i=1}^d B_{i,j}$ dimensional subspace where $B_{i,j} = 1$. Similarly, for VBC-Box, if $\phi(u)_i^\diamond \in \{0,1\}$ we can think of this determining the dimensions $D_u := \{i : \phi(u)_i^\diamond = 1\}$ which the box will be projected in. Given an edge $(u,v)$, we project into dimensions $D_u \cap D_v$ before determining the edge existence.

## 4.3 Representational Capacity

The energy functions for GBC-Box and VBC-Box were constructed such that Theorem 3 would not apply, thus making it possible that they may be capable of representing some graphs with cycles. In this section, we prove that both can model *any* directed graph.

**Theorem 5.** *Given a directed graph $G = (\mathcal{V}, \mathcal{E})$ and any $\tau, \nu \geq 0$ and $k \geq 2$, there exists a threshold $T > 0$, parameters $\phi : \mathcal{V} \to \mathbb{R}^{2d}$, and binary codes $B \in [0,1]^{d \times k}$ for which*

$$\text{E}_{\text{GBC-Box}}(u,v;(\phi,B),(\tau,\nu)) < T \quad \Longleftrightarrow \quad (u,v) \in \mathcal{E}.$$

*Proof.* Given a directed graph $G = (\mathcal{V}, \mathcal{E})$, let $(<, \mathcal{V})$ be an arbitrary strict total order on the vertices. Then define subgraphs $D_1 = (\mathcal{V}, \mathcal{E}_1)$ and $D_2 = (\mathcal{V}, \mathcal{E}_2)$ where $\mathcal{E}_1 = \{(u,v) \mid u < v\} \cap \mathcal{E}$ and $\mathcal{E}_2 = \{(u,v) \mid u > v\} \cap \mathcal{E}$. Observe that $D_1$ and $D_2$ are directed acyclic graphs, and thus Corollary 2 implies that for $j \in \{1,2\}$ there exists a threshold $T_j$, dimension $d_j = \mathcal{O}((\Delta + 2)\log|\mathcal{V}|)$, and mapping $\phi_j : \mathcal{V} \to \mathbb{R}^{2d_j}$ such that

$$\text{E}_{\text{G-Box}}(u,v;\phi,(\tau,\nu)) < T_j \quad \Longleftrightarrow \quad (u,v) \in \mathcal{E}_j.$$

Let $d = d_1 + d_2$, $T = \min(T_1, T_2)$, and define $\phi : V \to \mathbb{R}^{2d}$ and $B \in [0,1]^{d \times k}$ as follows:

$$\forall i \in \{1, \ldots, d_1\}, \quad \phi(u)_i^\pm = \phi_1(u)_i^\pm, \quad B_{i,1} = \tfrac{T}{T_1}, \quad B_{i,2} = 0,$$

$$\forall i \in \{d_1 + 1, \ldots, d_1 + d_2\}, \quad \phi(u)_i^\pm = \phi_2(u)_i^\pm, \quad B_{i,1} = 0, \quad B_{i,2} = \tfrac{T}{T_2}.$$

Then we have $\text{E}_{\text{GBC-Box}}(u,v;(\phi,B),(\tau,\nu)) = \min_{j \in \{1,2\}} \tfrac{T}{T_j} \text{E}_{\text{G-Box}}(u,v;\phi_j,(\tau,\nu))$ which completes the proof with $k = 2$, and therefore implies the result for all $k > 2$. $\square$

While motivated by a similar idea, note that VBC-BOX is not a generalization of GBC-BOX, and thus an independent proof of representational capacity is required.

**Theorem 6.** *Given a directed graph $G = (\mathcal{V}, \mathcal{E})$ and any $\tau, \nu \geq 0$, there exists a threshold $T > 0$ and parameters $\phi : \mathcal{V} \to \mathbb{R}^{2d} \times [0,1]^d$ for which $\mathrm{E}_{\text{VBC-BOX}}(u, v; \phi, (\tau, \nu)) < T$ if and only if $(u, v) \in \mathcal{E}$.*

*Proof.* Given a graph $G = (\mathcal{V}, \mathcal{E})$ let $H = \{\{u, v\} \mid u, v \in \mathcal{V}, u \neq v\}$. We will construct a VBC-BOX model in $d = |H|$ dimensions. For convenience, index the dimensions using $h \in H$. Then let $\phi(u)_h^\diamond = 1$ if $u \in h$, and 0 otherwise. This means when evaluating the edge $(u, v)$ or $(v, u)$ we simply need to compare in the 1-d space obtained by projecting the boxes to dimension $h = \{u, v\}$, and furthermore that this dimension will not be used when considering any other edges. Any directed graph on 2 nodes can be embedded using boxes in 1-dimension (observable by direct construction), which completes the proof. $\qquad\square$

While Theorem 5 and Theorem 6 are helpful in establishing that, unlike all prior box embedding models and many other geometric embeddings, GBC-BOX and VBC-BOX are capable of modeling any directed graph, the implied dimensionality bounds are far from optimal. In general, both models tend to require fewer dimensions than alternatives, which we analyze theoretically in Appendix D and observe empirically in Section 5.

### 4.4 Learning

We fit geometric embeddings by optimizing a binary cross entropy objective. Given some edges in a training set $\mathcal{T}$, the loss is defined as

$$L_{BCE}(\phi; \boldsymbol{\lambda}) = \sum_{(u,v) \in \mathcal{T}} \left[ \mathrm{E}(u, v; \phi, \boldsymbol{\lambda}) - \sum_{(u',v') \in N(u,v)} \log \left( 1 - e^{-\mathrm{E}(u',v';\phi,\boldsymbol{\lambda})} \right) \right] \quad (10)$$

where $N(u, v)$ is the set of negative samples for each positive edge $(u, v)$ within one batch. We sample minibatches of positive edges in $\mathcal{T}$, and for each positive edge we sample 32 edges not in $\mathcal{T}$ by randomly corrupting either the source or target node. We also use a self-adversarial negative weight, as described in [26].

### 4.5 Limitations and Regularization

There are a few limitations of Binary Code Box Embeddings: 1) **Transitivity and Flexibility** In order to model cycles with separate sub-spaces, the inductive bias of asymmetric transitivity might be weakened. 2) **Inefficiency of Parameterization** For VBC-BOX, the number of parameters is increased by 50% compared with a G-BOX model in equal dimension. In addition, during inference, a large portion of box co-ordinates are "dead" when binary codes are near zero. Here, we introduce a regularization method and a tunable binary code size to resolve these concerns.

**Regularization** We can regularize the sparsity of binary codes to penalize dimension drop-off using the lasso with a regularization weight $w_r$, leading to a loss function $L = L_{BCE} + w_r \|1 - B\|_{\ell^1}$.

**Restricted Binary Code Size** We can constrain the number of trainable binary codes to the last $d_{\text{bin}}$ dimensions, setting the first $d - d_{\text{bin}}$ dimensions to 1.

Both $w_r$ and $d_{\text{bin}}$ can provide a handle on mitigating issues mentioned above. One can increase $w_r$ or decrease $d_{\text{bin}}$ to preserve more transitivity and make full use of the model's parameters.

## 5 Experiments

### 5.1 Graph Reconstruction

While GBC-BOX and VBC-BOX can provably model any directed graph, the extent to which they can be trained to do so via gradient descent is another matter. In this section, we compare the reconstruction performance of various geometric embedding methods on a number of synthetic graphs, including simple directed cycles, trees and scale-free networks.

**Baselines.** We compare our model with different baselines:

*Vector**: We implement a vector baseline where each node is parameterized by a source and target vector, and the energy function is measured by $\mathrm{E}(u,v) = -\log \sigma(\phi(u)_{\text{source}} \cdot \phi(v)_{\text{target}})$. * indicates it uses source and target vectors.

*Lorentzian*: It has been shown that hyperbolic space can embed undirected trees with arbitrarily low distortion [22], therefore we also compare with the baseline of squared Lorentzian distance on the hyperboloid [10, 2].

*Hyperbolic Entailment Cones*: Ganea et al. [7] model vertices as cones in hyperbolic space, combining the bias of hyperbolic space to represent tree-like graphs with the transitivity bias of region-based representations.

*Box Models*: We also compare with G-Box [4] and T-Box [2], as defined in Section 2.2.

We use Bayesian hyperparameter tuning based on the optimal F1 score for reconstruction.

**Capacity over Cycles.** We evaluate each model's capacity to represent cycles, where simple directed cycles are generated with an increasing number of vertices ($|\mathcal{V}| = 2^2, 2^4, 2^8, 2^{12}$). In addition, we analyze the effect of adding one reverse edge to the cycle, which conventional box embedding models can represent. Results are shown in Table 1. For a fair comparison, all methods use 12 parameters per vertex [6]. VBC-Box shows the best reconstruction performance. Most other geometric baselines cannot model cycles. Surprisingly, VBC-Box even outperforms the *Vector** baseline when $|\mathcal{V}| = 2^{12}$, indicating our model's high expressivity and surprising ease of training. Results also show that, in concordance with Proposition 1, G-Box can model a cycle containing a reverse edge when $|\mathcal{V}|$ is small, though it may struggle to do so. We also see that T-Box can model certain cycles when $|\mathcal{V}|$ is small, which is not a contradiction of Theorem 3 as the energy function for T-Box does not satisfy the premise.

Table 1: Reconstruction performance (F1 score) on directed cycles. All embeddings use 12 parameters per vertex. Different columns show the results as we increase the number of vertices in the cycle.

| Methods | Simple cycle | | | | + One bidirectional edge | | | |
|---|---|---|---|---|---|---|---|---|
| $|\mathcal{V}| =$ | $2^2$ | $2^4$ | $2^8$ | $2^{12}$ | $2^2$ | $2^4$ | $2^8$ | $2^{12}$ |
| Vector* | **1.0** | **1.0** | **1.0** | 0.676 | **1.0** | **1.0** | **1.0** | 0.666 |
| Lorentzian | 0.857 | 0.75 | 0.679 | 0.671 | **1.0** | 0.839 | 0.693 | 0.665 |
| Hyperbolic Entailment Cones | 0.75 | 0.667 | 0.662 | 0.635 | 0.75 | 0.692 | 0.654 | 0.645 |
| G-Box | 0.857 | 0.762 | 0.695 | 0.648 | **1.0** | 0.914 | 0.689 | 0.630 |
| T-Box | **1.0** | **1.0** | 0.996 | 0.685 | **1.0** | **1.0** | 0.992 | 0.659 |
| GBC-Box | **1.0** | **1.0** | 0.992 | 0.957 | **1.0** | **1.0** | 0.993 | 0.967 |
| VBC-Box ($d_{bin} = d$) | **1.0** | **1.0** | **1.0** | **0.973** | **1.0** | **1.0** | **1.0** | **0.978** |

**Capacity over Trees.** It is known that box models have an inductive bias which makes them more suited to capturing asymmetric transitivity, whereas hyperbolic space is particularly suitable for undirected trees. In this experiment, we evaluate whether binary code models maintain the inductive bias of box embeddings. We generated four balanced (out-)trees, each with $2^{13}$ vertices and branching factor of $[2, 3, 5, 10]$. We also generated another four graphs with full transitive closures. We compare each method with 12, 24 and 48 parameters per vertex. For each setting, we average the results over the four trees, and present the results in Table 2. As expected, G-Box performs the best on transitively closed trees, while the Lorentzian model performs well on balanced trees. It is shown that GBC-Box performs equally well as G-Box on transitively closed trees, while the performance of VBC-Box is slightly lower. In contrast, the latter performs similarly or better than the former on the transitive reduction of trees. This suggests that more transitive bias is preserved in GBC-Box, while per vertex binary codes provide more representational flexibility. In addition, these results confirm a trend observed in [2], where binary code models and and T-Box outperform the Lorentzian model on balanced trees when dimension size is increased, and the performance of other geometric-based embeddings are not capable of overfitting even as dimension increases.

**Capacity over Random Graphs.** Finally, we conduct experiments on scale-free networks, a simulation to real-world graphs, where the edge distribution follows preferential attachment. In order to analyze how the cyclicity of graphs affects each model's performance, we randomly sampled

---

[6]12 is the least common multiple of 2, 3, 4 which are the minimum number of parameters per node for G-Box, T-Box and VBC-Box

Table 2: Average reconstruction performance (F1 score) on balanced trees with $|\mathcal{V}| = 2^{13}$ and branching factors of $2, 3, 5, 10$ using $12, 24$, and $48$ parameters per vertex.

| Methods | Balanced tree | | | w. transitive closures | | |
|---|---|---|---|---|---|---|
| # Parameters / vertex = | 12 | 24 | 48 | 12 | 24 | 48 |
| Vector* | 0.453 | 0.992 | **1.0** | 0.863 | 0.999 | **1.0** |
| Lorentzian | **0.929** | 0.935 | 0.951 | 0.975 | 0.979 | 0.995 |
| Hyperbolic Entailment Cones | 0.828 | 0.834 | 0.838 | 0.977 | 0.982 | 0.987 |
| G-Box | 0.832 | 0.830 | 0.842 | **1.0** | **1.0** | **1.0** |
| T-Box | 0.800 | 0.957 | **1.0** | 0.952 | 0.997 | **1.0** |
| GBC-Box | 0.901 | 0.961 | 0.983 | **1.0** | **1.0** | **1.0** |
| VBC-Box ($d_{bin} = d$) | 0.866 | **0.994** | **1.0** | 0.987 | 0.999 | **1.0** |

nearly three-thousand graphs using a wide range of parameters used for graph generation. Then we split the generated graphs into five bins by our proposed measure *cyclicity*: The proportion of vertices in a given graph involved in at least one cycle. In order to analyze models' effectiveness of modeling cycles instead of the density of graphs, we randomly sample 5 graphs from each bin where the average degree is in the range between 3 and 4. Results are shown in Figure 2. From the chart, we can see that our proposed model outperforms standard G-Box in all scenarios. Moreover, the gap in performance is more significant when cyclicity is high. Figure 2 also clearly shows that VBC-Box provides more representational capacity overall, while GBC-Box is less expressive in modeling cyclic graphs.

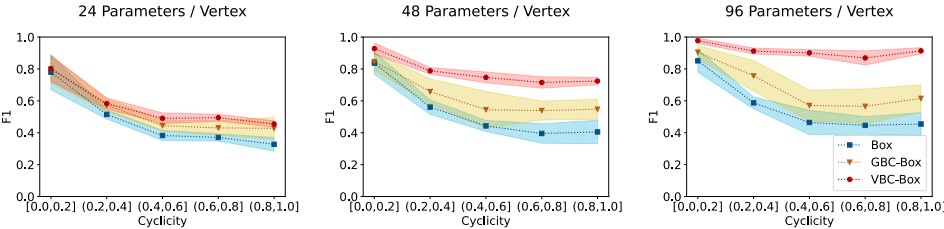

Figure 2: Reconstruction performance on Scale-free networks with $|\mathcal{V}| = 2^{13}$. We plot the F1 scores for G-Box, GBC-Box and VBC-Box using 24, 48, 96 parameters per vertex.

## 5.2 Link Prediction

Finally, we apply our binary code box models on link prediction tasks to evaluate the models' generalization ability. We evaluate on the following real world graphs: Google hyperlink network, Epinions trust network, CORA citation network, Twitter network, and DREAM5 gene regulatory networks. For more data statistics, see Appendix E. During training, all hyperparameters are tuned via 10% cross-validation and the test set results are averaged over 10 models of different random seeds, which were trained on the full training set with the best hyper-parameters found during cross-validation. We also tune the weight of the sparsity regularization $w_r$ for all BC-Box models (see Section 4.5).

In Table 3, we follow [30] and compare with several baselines including vector-based baselines such as DeepWalk [17], LINE [27], node2vec [8], HOPE [15], APP [29], DGGAN [30] and our own implementation of a vector-based model with separate source and target parameters, as well as region-based baselines such as G-Box [3] and T-Box [2]. Models are evaluated using the Area Under ROC Curve (AUC). For fair comparison, we follow [30] and use 128 parameters per vertex for all our models. Results show that our binary code box models out-perform other baselines in most cases. Furthermore, there is an clear trend that VBC-Box performs the best when graphs are highly cyclic (on the left side of the table), whereas VBC-Box with fewer binary code dimensions and GBC-Box model start to perform better when graphs are less cyclic. In the case where a graph is mostly acyclic (on the right side of the table), G-Box performs equally well.

We also observe that box geometry is superior than vector-based models in all scenarios, even if graphs are less transitive, or have a lot of cycles, showing the strength of box geometry in modeling

directed graphs. In addition, we also compare with another recent work from Sim et al. [23] over DREAM5 datasets, where we observe that box embedding models out-performs their baselines significantly in most cases. (On *In Silico* dataset, our model has an average precision of 68.8%, out-performs their best result 61.0%.) Detailed results are in Appendix F.

Table 3: **Link prediction on real-world graphs** We use AUC as evaluation metric. Vector-based methods (upper), and box embedding variants (bottom). Most vector baseline results are provided by [30]. We evaluated over two negative sampling strategies for testing, *unif.*: uniformly sampled negatives; *corr.*: randomly corrupting source or target node in each positive edge in the test set. All methods use 128 parameters per vertex. Bold numbers perform the best, and underscored numbers perform the best in all non-box models

| Methods | Google | | Epinions | | CORA | | Twitter | |
| --- | --- | --- | --- | --- | --- | --- | --- | --- |
| *Cyclicity =* | *0.96* | | *0.88* | | *0.23* | | *0.01* | |
| *Transitivity =* | *0.40* | | *0.09* | | *0.22* | | *0.01* | |
| | unif. | corr. | unif. | corr. | unif. | corr. | unif. | corr. |
| DeepWalk [30] | 83.6 | - | 76.6 | - | 84.9 | - | 50.4 | - |
| LINE-1 [30] | 89.7 | - | 78.8 | - | 84.7 | - | 53.1 | - |
| node2vec [30] | 84.3 | - | 89.7 | - | 85.3 | - | 50.6 | - |
| HOPE* [30] | 87.5 | - | 79.6 | - | 77.6 | - | 98.0 | - |
| APP* [30] | 92.1 | - | 70.5 | - | 76.6 | - | 71.6 | - |
| DGGAN [30] | 92.3 | - | 96.1 | - | 85.1 | - | 99.7 | - |
| Vector(Ours)* | 94.0 | 94.2 | 93.0 | 88.9 | 78.9 | 76.7 | 99.8 | 84.1 |
| G-Box | 99.2 | 98.2 | 95.1 | 90.0 | 93.9 | 89.6 | 99.8 | **86.3** |
| T-Box | 97.1 | 95.8 | 96.4 | 89.6 | 87.3 | 79.6 | 99.8 | 84.4 |
| GBC-Box | 99.0 | 98.3 | 96.8 | **92.7** | 92.7 | **90.3** | **99.9** | 86.2 |
| VBC-Box ($0 < d_{bin} < d$) | 99.3 | 98.3 | 97.6 | 92.2 | **94.1** | 89.7 | **99.9** | 86.1 |
| VBC-Box ($d_{bin} = d$) | **99.5** | **98.6** | **98.0** | 92.4 | 93.2 | 88.8 | 99.8 | 85.7 |

# 6 Conclusion

In this paper, we introduced binary code box embeddings, a generalized box embedding method to model directed graphs. We provide both theoretical and empirical results showing the capacity of our model for modeling directed graphs. We demonstrated that this model can maintain a useful bias of transitivity while also modeling graphs with cycles.

### Acknowledgements

This work was supported in part by the Center for Data Science and the Center for Intelligent Information Retrieval, and in part by the National Science Foundation under Grants No. 2106391 and No. 1763618, and in part by IBM Research AI through AI Horizons Network, and in part by the Chan Zuckerberg Initiative under the project "Scientific Knowledge Base Construction", and the Office of Naval Research (ONR) via Contract No. N660011924032 under Subaward No. 123875727 from the University of Southern California. The U.S. Government is authorized to reproduce and distribute reprints for Governmental purposes notwithstanding any copyright notation thereon. Some of the work reported here was performed using high performance computing equipment obtained under a grant from the Collaborative R&D Fund managed by the Massachusetts Technology Collaborative. Any opinions, findings and conclusions or recommendations expressed in this material are those of the authors and do not necessarily reflect those of the sponsor.

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
