## A  Equivalence of G-Box and Dasgupta et al. [4]

In our notation, the model in Dasgupta et al. [4] would have score function

$$F_{\text{D-Box}}((x^-, x^+), (y^-, y^+); (\tau, \nu)) \coloneqq \frac{\text{LSE}_\nu \left(-\text{LSE}_\tau(-x^+, -y^+) - \text{LSE}_\tau(x^-, y^-) - 2\nu\gamma, 0\right)}{\text{LSE}_\nu(y^+ - y^- - 2\nu\gamma, 0)}$$

where $\gamma$ is the Euler-Mascheroni constant.

As presented in Dasgupta et al. [4], D-Box used a single temperature $\beta = \tau = \nu$, and derived this score function as an approximation to a ratio of expected volumes of intervals whose endpoints were modeled by Gumbel random variables. Gumbel random variables are typically parameterized by a *location* and *scale*, and Dasgupta et al. [4] interpreted the parameters $x^-, y^-$ (resp. $x^+, y^+$) as the location parameters for GumbelMax (resp. GumbelMin) distributions with scale $\beta$.

**Remark 1.** *Although the model was proposed and analyzed in Dasgupta et al. [4] using a single temperature parameter $\beta$, the authors do use separate $\tau$ and $\nu$ parameters when implementing the model, and so we adopt that formulation when defining $F_{\text{D-Box}}$ above.*

We claim D-Box and G-Box are equivalent, in the following sense.

**Proposition 2.** *Given any $\nu \geq 0$ there exists a bijection $\psi$ on the set of functions $\{V \to \mathbb{R}^{2d}\}$ such that*

$$E_{\text{G-Box}}(u, v; \psi(\phi), (\tau, \nu)) = E_{\text{D-Box}}(u, v; \phi, (\tau, \nu)), \tag{11}$$

*and, being a bijection,*

$$E_{\text{G-Box}}(u, v; \phi, (\tau, \nu)) = E_{\text{D-Box}}(u, v; \psi^{-1}(\phi), (\tau, \nu)). \tag{12}$$

*Proof.* Observe that for any $a, b, c \in \mathbb{R}$ and $t \geq 0$,

$$\text{LSE}_t(a + c, b + c) = \text{LSE}_t(a, b) + c.$$

Then

$$F_{\text{G-Box}}((x^- + \nu\gamma, x^+ - \nu\gamma), (y^- + \nu\gamma, y^+ - \nu\gamma); (\tau, \nu))$$
$$= \frac{\text{LSE}_\nu \left(-\text{LSE}_\tau(-x^+ + \nu\gamma, -y^+ + \nu\gamma) - \text{LSE}_\tau(x^- + \nu\gamma, y^- + \nu\gamma), 0\right)}{\text{LSE}_\nu(y^+ - \nu\gamma - y^- - \nu\gamma, 0)}$$
$$= \frac{\text{LSE}_\nu \left(-\text{LSE}_\tau(-x^+, -y^+) - \text{LSE}_\tau(x^-, y^-) - 2\nu\gamma, 0\right)}{\text{LSE}_\nu(y^+ - y^- - 2\nu\gamma, 0)}$$
$$= F_{\text{D-Box}}((x^-, x^+), (y^-, y^+); (\tau, \nu)).$$

Therefore, as introduced in Section 2, if we label the output of $\phi$ using $d$ pairs $\phi(u)_i = (\phi(u)_i^-, \phi(u)_i^+)$ and define $\psi(\phi)$ to be a mapping from $V \to \mathbb{R}^{2d}$ such that

$$\psi(\phi)(u)_i = (\phi(u)_i^- + \nu\gamma, \ \phi(u)_i^+ - \nu\gamma), \tag{13}$$

the calculations above prove (11), and the proof of (12) is similar. $\square$

**Remark 2.** *Note that the mean of $X \sim \text{GumbelMax}(\mu, \beta)$ is $\mu + \beta\gamma$, and similarly the mean of $X \sim \text{GumbelMin}(\mu, \beta)$ is $\mu - \beta\gamma$. As mentioned above, in the setting where $\beta = \tau = \nu$ the parameters of D-Box can be interpreted as the location parameters for Gumbel distributions, and thus (13) simply takes the location parameters to their mean. Hence, in the case where $\tau = \nu$, the G-Box model can simply be interpreted as using the mean of the Gumbel distributions as opposed to the location parameter. This leads to a slight simplification in the score function by removing the $2\nu\gamma$, which has minor computational and mathematical benefits. The more conceptual benefit, however, is that it unifies Box, G-Box, and T-Box.*

## B  Representational Capacity of G-Box

In this section we prove any graph capable of being represented by Box is also capable of being represented by G-Box, regardless of the temperature hyperparameters. The proof involves two components:

1. The representational capacity of G-Box depends not on the absolute values of $\tau$ and $\nu$, but rather their ratio.

2. The energy of Box can be approximated by G-Box using small enough $\tau$ and $\nu$.

**Proposition 3.** *Let $\tau_1, \nu_1 > 0$ and $\phi : V \to \mathbb{R}^{2d}$ be given. Then for any $\tau_2, \nu_2 > 0$ such that $\frac{\tau_2}{\nu_2} = \frac{\tau_1}{\nu_1}$ the function $\psi(\phi) : V \to \mathbb{R}^{2d}$ for which $\psi(\phi)(u)_i^{\pm} = \frac{\nu_2}{\nu_1}\phi(u)_i^{\pm}$ is such that*

$$\mathrm{E}_{\text{G-Box}}(u, v; \phi, (\tau_1, \nu_1)) = \mathrm{E}_{\text{G-Box}}(u, v; \psi(\phi), (\tau_2, \nu_2)). \tag{14}$$

*Proof.* The proof is by direct calculation. First, note that for any $t, c > 0$ we have for any vector $\mathbf{x} \in \mathbb{R}^n$

$$\mathrm{LSE}_t(c\mathbf{x}) = t \log\left(\sum_{i=1}^{n} \exp\left(\tfrac{cx_i}{t}\right)\right) = c(t/c) \log\left(\sum_{i=1}^{n} \exp\left(\tfrac{x_i}{t/c}\right)\right) = c\,\mathrm{LSE}_{t/c}(\mathbf{x}).$$

In particular, for $c = \frac{\nu_2}{\nu_1} = \frac{\tau_2}{\tau_1}$ (where the latter equality follows from the premise of the proposition) we have for any $a, b \in \mathbb{R}$

$$\mathrm{LSE}_{\tau_2}(ca, cb) = c\,\mathrm{LSE}_{\tau_1}(a, b) \quad \text{and} \quad \mathrm{LSE}_{\nu_2}(ca - cb, 0) = c\,\mathrm{LSE}_{\nu_1}(a - b, 0).$$

Thus

$$
\begin{aligned}
&F_{\text{G-Box}}((cx^-, cx^+), (cy^-, cy^+); (\tau_2, \nu_2)) \\
&\quad = \frac{\mathrm{LSE}_{\nu_2}\left(-\mathrm{LSE}_{\tau_2}(-cx^+, -cy^+) - \mathrm{LSE}_{\tau_2}(cx^-, cy^-), 0\right)}{\mathrm{LSE}_{\nu_2}(cy^+ - cy^-, 0)} \\
&\quad = \frac{\mathrm{LSE}_{\nu_2}\left(-c\,\mathrm{LSE}_{\tau_1}(-x^+, -y^+) - c\,\mathrm{LSE}_{\tau_1}(x^-, y^-), 0\right)}{\mathrm{LSE}_{\nu_2}(cy^+ - cy^-, 0)} \\
&\quad = \frac{c\,\mathrm{LSE}_{\nu_1}\left(-\mathrm{LSE}_{\tau_1}(-x^+, -y^+) - \mathrm{LSE}_{\tau_1}(x^-, y^-), 0\right)}{c\,\mathrm{LSE}_{\nu_1}(y^+ - y^-, 0)} \\
&\quad = F_{\text{G-Box}}((x^-, x^+), (y^-, y^+); (\tau_1, \nu_1)),
\end{aligned}
$$

which proves (14).

The following lemma will be helpful in proving the next part.

**Lemma 1.** *For all $y > 0$, given $\varepsilon > 0$ and some $M \in \mathbb{R}$, there exists $\delta > 0$ such that for all $0 < \nu < \delta$, for all $x < M$ we have*

$$\left| \frac{\mathrm{LSE}_\nu(x, 0)}{\mathrm{LSE}_\nu(y, 0)} - \frac{\max(x, 0)}{y} \right| < \varepsilon.$$

*Proof.* Note that $\mathrm{LSE}_\nu(x, 0)$ is monotonically increasing in $x$ for any $\nu \geq 0$, and is always greater than $\max(x, 0)$. Furthermore,

$$|\mathrm{LSE}_\nu(x, 0) - \max(x, 0)| = \mathrm{LSE}_\nu(x, 0) - \max(x, 0) \leq \nu \log 2$$

as it obtains it's maximum when $x = 0$ (which can be observed by inspection of the signs of the derivative). Then for all $x < M$ we have

$$
\left| \frac{\mathrm{LSE}_\nu(x, 0)}{\mathrm{LSE}_\nu(y, 0)} - \frac{\max(x, 0)}{y} \right| < \left| \frac{\mathrm{LSE}_\nu(x, 0)}{\mathrm{LSE}_\nu(y, 0)} - \frac{\mathrm{LSE}_\nu(x, 0)}{y} \right| + \left| \frac{\mathrm{LSE}_\nu(x, 0)}{y} - \frac{\max(x, 0)}{y} \right|.
$$

$$
< \mathrm{LSE}_\nu(M, 0)\left| \frac{1}{\mathrm{LSE}_\nu(y, 0)} - \frac{1}{y} \right| + \frac{\nu \log 2}{y}. \tag{15}
$$

Now (15) does not depend on $x$, and tends to 0 as $\nu \to 0$, which completes the proof. $\qquad\square$

**Proposition 4.** *Given a mapping $\phi : V \to \mathbb{R}^{2d}$ where $\phi(u)_i^+ > \phi(u)_i^-$ for each $u \in V$, $i \in \{1, \ldots, d\}$, we have that for all $u, v \in V$,*

$$\lim_{(\tau, \nu) \to (0,0)} \mathrm{E}_{\text{G-Box}}(u, v; \phi, (\tau, \nu)) = \mathrm{E}_{\text{Box}}(u, v; \phi). \tag{16}$$

*Proof.* Given fixed $x^- < x^+, y^- < y^+$, let $f(\tau) = -\text{LSE}_\tau(-x^+, -y^+) - \text{LSE}(x^-, y^-)$, and $z = \min(x^+, y^+) - \max(x^-, y^-) = \lim_{\tau \to 0} f(\tau)$. Then

$$\left| F_{\text{G-Box}}((x^-, x^+), (y^-, y^+); (\tau, \nu)) - F_{\text{Box}}((x^-, x^+), (y^-, y^+)) \right|$$

$$= \left| \frac{\text{LSE}_\nu(f(\tau), 0)}{\text{LSE}_\nu(y^+ - y^-, 0)} - \frac{\max(z, 0)}{y^+ - y^-} \right|$$

$$< \left| \frac{\text{LSE}_\nu(f(\tau), 0)}{\text{LSE}_\nu(y^+ - y^-, 0)} - \frac{\max(f(\tau), 0)}{y^+ - y^-} \right| + \left| \frac{\max(f(\tau), 0)}{y^+ - y^-} - \frac{\max(z, 0)}{y^+ - y^-} \right|. \quad (17)$$

Given $\varepsilon > 0$, choose $\delta_1$ such that $0 < \tau < \delta_1$ implies the second summand in (17) is bounded by $\varepsilon/2$. Then $f(\tau)$ is bounded, and we can apply Lemma 1 to choose $\delta_2$ such that $0 < \nu < \delta_2$ implies the first summand is less than $\varepsilon/2$. Thus taking $\delta = \min(\delta_1, \delta_2)$ completes the proof on the level of the per-dimension score functions, and thus (16) follows by continuity. $\quad\square$

We are now ready to prove the main theorem.

$\square$

**Theorem 1.** *Given a threshold $T$, temperature hyperparameters $\tau, \nu$, there exists and a bijection $\psi$ on the set of parameterizations $\{\mathcal{V} \to \mathbb{R}^{2d}\}$ such that for all $u, v \in \mathcal{V}$,*

$$\text{E}_{\text{G-Box}}(u, v; \psi(\phi), (\tau, \nu)) < T \quad \Longleftrightarrow \quad \text{E}_{\text{Box}}(u, v; \phi) < T.$$

*Proof.* Let $\varepsilon > 0$ be a number we will specify later. Then by Proposition 4, for each $(u, v) \in V^2$ we have some $\delta_{(u,v)} > 0$ such that

$$\tau, \nu \in (0, \delta_{(u,v)}) \quad \Longrightarrow \quad |\text{E}_{\text{G-Box}}(u, v; \phi, \tau, \nu) - \text{E}_{\text{Box}}(u, v; \phi)| < \varepsilon. \quad (18)$$

Let

$$\delta = \min_{(u,v) \in V^2} \delta_{(u,v)}, \quad \tau' = \frac{\delta\tau}{2\max(\tau, \nu)}, \quad \nu' = \frac{\delta\nu}{2\max(\tau, \nu)}.$$

Since $\frac{\tau'}{\nu'} = \frac{\tau}{\nu}$ we can apply Proposition 3, which guarantees the existence of a function $\psi$ such that

$$\text{E}_{\text{G-Box}}(u, v; \psi(\phi), (\tau', \nu')) = \text{E}_{\text{Box}}(u, v; \phi, (\tau, \nu)).$$

Noting that $\tau', \nu' \in (0, \delta)$, we can combine this with (18), and find

$$\text{E}_{\text{Box}}(u, v; \phi) - \varepsilon < \text{E}_{\text{G-Box}}(u, v; \psi(\phi), (\tau, \nu)) < \text{E}_{\text{Box}}(u, v; \phi) + \varepsilon.$$

Let

$$T_1 = \max_{(u,v) \in \mathcal{E}} \text{E}_{\text{Box}}(u, v; \phi), \quad T_2 = \min_{(u,v) \notin \mathcal{E}} \text{E}_{\text{Box}}(u, v; \phi),$$

and set $\varepsilon = \min(T - T_1, T_2 - T)$. Then if $(u, v) \in \mathcal{E}$ we have

$$\text{E}_{\text{G-Box}}(u, v; \psi(\phi), (\tau, \nu)) < T_1 + T - T_1 = T,$$

and if $(u, v) \notin \mathcal{E}$ we have

$$\text{E}_{\text{G-Box}}(u, v; \psi(\phi), (\tau, \nu)) > T_2 - (T_2 - T) = T,$$

which completes the proof. $\quad\square$

## C  Representing Cycles with Box Embeddings

**Proposition 1** *If $G$ is a directed graph which is the union of a chordless cycle and one reverse edge, G-Box can model $G$ in 2 dimensions.*

*Proof.* Given a $G = \{\mathcal{V}, \mathcal{E}\}$, $\mathcal{E} = \{(1, 2), (2, 3), (3, 4), (4, 5), , (N-1, N), (N, 1)\} \cup \{(1, N)\}$ When $N = 2$, it is trivial since two boxes can be equal or overlap with each other largely (as shown in Fig 3a). When $N = 3$, we can construct boxes as shown in Fig 3b, where $\text{E}_{\text{Box}}(1, 2) = -\log 0.5$, $\text{E}_{\text{Box}}(2, 3) = -\log 1/3$, $\text{E}_{\text{Box}}(3, 1) = -\log 0.375$, $\text{E}_{\text{Box}}(2, 1) = -\log 0.25$, $\text{E}_{\text{Box}}(3, 2) =$

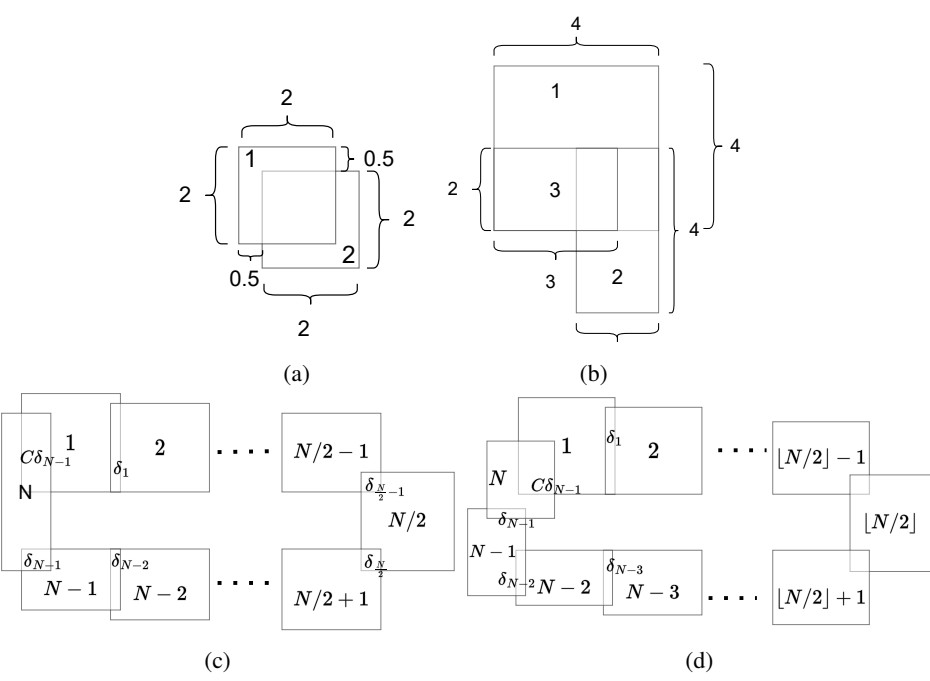

Figure 3: Visualization of 2D box to represent a graph with chordless cycles and one reverse edge. Diagram 3a can represent a 2-node cycle ($1 \leftrightarrow 2$). Diagram 3b can represent a 3-node cycle with a reverse edge ($1 \to 2 \to 3 \leftrightarrow 1$). Diagram 3c is when there are even number of nodes in the graph. And diagram 3d is is when the number is odd.

$-\log 0.25$, $\mathrm{E_{Box}}(1,3) = -\log 1.0$. Therefore if $T = -\log 0.3$, they can represent a graph $1 \to 2 \to 3 \leftrightarrow 1$.

When $N > 3$ and $N$ is an even number, we can construct a 2-d BOX as in Figure 3c. Let the area of $\phi(i)$ be $V(\phi(i))$, the area of intersection box between two nodes that are connected is $\delta_i = V(\phi(i) \cap \phi(i+1))$, $i \le N - 1$, and let $V(\phi(N) \cap \phi(1)) = C\delta_{N-1}$. We align the boxes as shown in Figure 3c with constant width and decreasing heights such that $V(i+1) = \alpha V(i)$ for some value $\alpha$. Then for large enough $\alpha$, there exists arrangement of boxes as shown in Figure 3c such that $\delta_{i+1} = \alpha\delta_i$, $i \le N - 2$ and $C = \frac{1}{\alpha^{N-1}}$. Apart from boxes $N/2$ and $N$, this is straightforward, as their decreased heights will already amount to multiplying the size of the intersection by $\alpha$. For the boxes on the endpoints, we leverage the fact that we are not constrained to a fixed width to accomplish this. Then for $T = -\log \frac{\delta_1}{V(\phi(1))}$ we have

$$\forall i \in \{1, \dots, N-1\}, \quad \mathrm{E_{Box}}(i, i+1) = -\log \frac{\delta_1}{\alpha V(\phi(1))} < T$$

$$\mathrm{E_{Box}}(N, 1) = -\log \frac{\delta_1}{\alpha V(\phi(1))} < T \quad \mathrm{E_{Box}}(1, N) = -\log \frac{\delta_1}{\alpha^N V(\phi(1))} < T,$$

and for $(u, v) \notin \mathcal{E}$, $\mathrm{E_{Box}}(u, v) \ge T$. The argument for $N$ odd is similar, using an arrangement as shown in Figure 3d.

Given Appendix B, this proof also applies to G-BOX. $\qquad\square$

## D   Dimensionality Bounds for Binary Code Models

We start by proving the following lemma, which implies that any directed graph on 2 nodes can be embedded using boxes in 1-dimension with any threshold $T > 0$.

**Lemma 2.** *Given $T > 0$, there exist $x^-, x^+, y^-, y^+ \in \mathbb{R}$ such that*

$$-\log F_{\mathrm{Box}}((x^-, x^+), (y^-, y^+)) < T < -\log F_{\mathrm{Box}}((y^-, y^+), (x^-, x^+)).$$

*Proof.* The proof is a direct construction; take

$$y^- = x^- = 0, \quad x^+ = 1, \quad \text{and} \quad y^+ = e^{-2T}.$$

Then

$$\max(\min(1, e^{-2T}) - \max(0,0), 0) = e^{-2T},$$

and thus

$$F_{\text{Box}}((0,1), (0, e^{-T/2})) = 1, \quad \text{and} \quad F_{\text{Box}}((0, e^{-2T}), (0,1)) = e^{-2T}$$

which implies the desired result. □

We then strengthen the statement of Theorem 6 to apply to an arbitrary threshold.

**Theorem 7.** *Given a directed graph $G = (\mathcal{V}, \mathcal{E})$, any temperatures $\tau, \nu \geq 0$, and any threshold $T > 0$, there exists a parameterization $\phi : \mathcal{V} \to \mathbb{R}^{2d} \times [0,1]^d$ with $d = \mathcal{O}(|\mathcal{V}|^2)$ for which*

$$\mathrm{E}_{\text{VBC-Box}}(u, v; \phi, (\tau, \nu)) < T \quad \Longleftrightarrow \quad (u, v) \in \mathcal{E}.$$

*Proof.* Given a graph $G = (\mathcal{V}, \mathcal{E})$ let $H = \{\{u, v\} \mid u, v \in \mathcal{V}, u \neq v\}$. We will construct a VBC-Box model in $d = |H|$ dimensions. For convenience, index the dimensions using $h \in H$, and let $\phi(u)_h^\diamond = 0$ if $u \notin h$. Thus when evaluating edge $(u, v)$ or $(v, u)$, dimension $h = \{u, v\}$ is the only dimension whose score may not be equal to 1.

Lemma 2 implies any graph on 2 nodes can be embedded using threshold $T$ in one dimension, and Theorem 1 implies this is also true for G-Box for any setting of temperatures, completing the proof. □

**Theorem 8.** *Let $G = (\mathcal{V}, \mathcal{E})$, and let $\tau, \nu \geq 0$ be given temperature hyperparameters. Let $\mathcal{V}_F$ be the minimum feedback vertex set, $\mathcal{E}_F = \mathcal{E} \cap \mathcal{V}_F^2$, and $G_F = (\mathcal{V}_F, \mathcal{E}_F)$. Then for any temperatures $\tau, \nu$ there exists a threshold $T > 0$ and a parameterization $\phi : \mathcal{V} \to \mathbb{R}^{2d} \times [0,1]^d$ such that*

$$\mathrm{E}_{\text{VBC-Box}}(u, v; \phi, (\tau, \nu)) < T \quad \Longleftrightarrow \quad (u, v) \in \mathcal{E},$$

*where $d = \mathcal{O}((\Delta_F + 2) \log(|\mathcal{V}_F|) + |\mathcal{V}_C|^2)$, with $\Delta_F$ the maximum degree of $G_F$, and*

$$\mathcal{V}_C = \{u \mid (u, v) \in \mathcal{E}, u \notin \mathcal{V}_F \text{ or } v \notin \mathcal{V}_F\}.$$

*Proof.* Theorem 2 implies that $G_F$ can be embedded using G-Box with the given temperature hyperparameters $\tau, \nu$ in dimension at most $d_F = \mathcal{O}((\Delta_F + 2) \log |\mathcal{V}_F|)$. Let $\phi_F : \mathcal{V}_F \to \mathbb{R}^{2d_F}$ be the parameterization for this embedding, and $T$ the threshold on the energy. Now let

$$\mathcal{E}_{\neg F} = \{(u, v) \in \mathcal{V}^2 \mid u \notin \mathcal{V}_F \quad \text{or} \quad v \notin \mathcal{V}_F\},$$

and define $\mathcal{E}_1 = \mathcal{E}_F \cup \mathcal{E}_{\neg F}$ and $G_1 = (\mathcal{V}, \mathcal{E}_1)$. We can extend $\phi_F$ to a VBC-Box parameterization $\phi_1 : \mathcal{V} \to \mathbb{R}^{2d_F} \times [0,1]^{2d_F}$ as follows:

$$\phi_1(u)_i = \begin{cases} (\phi_F(u)_i^-, \phi_F(u)_i^+, 1) & \text{if} \quad u \in \mathcal{V}_F, \\ (0, 1, 0) & \text{otherwise.} \end{cases}$$

This parameterization is such that

$$\mathrm{E}_{\text{VBC-Box}}(u, v; \phi_F, (\tau, \nu)) \begin{cases} = 0 & \text{if} \quad (u, v) \in \mathcal{E}_{\neg F}, \\ < T & \text{if} \quad (u, v) \in \mathcal{E}_F, \\ > T & \text{otherwise.} \end{cases} \tag{19}$$

Now let $\mathcal{E}_C = \mathcal{E} \cap \mathcal{E}_{\neg F}$, and note that these edges are only between nodes in $\mathcal{V}_C$. Define $G_C = (\mathcal{V}_C, \mathcal{E}_C)$, which, by Theorem 7, can be embedded with threshold $T$ in $d_C = \mathcal{O}(|\mathcal{V}_C|^2)$ dimensions. Let $\phi_C : \mathcal{V}_C \to \mathbb{R}^{2d_C}$ be the associated parameterization.

Now let $G_2 = (\mathcal{V}, \mathcal{E}_C \cup (\mathcal{V}_F \times \mathcal{V}_F))$, then extend $\phi_C$ to a parameterization $\phi_2 : \mathcal{V} \to \mathbb{R}^{2d_C}$ as follows:

$$\phi_2(u)_i = \begin{cases} (\phi_C(u)_i^-, \phi_C(u)_i^+, 1) & \text{if} \quad u \in \mathcal{V}_C, \\ (0, 1, 0) & \text{otherwise.} \end{cases}$$

For this parameterization, we have

$$\mathrm{E}_{\text{VBC-Box}}(u, v; \phi_2, (\tau, \nu)) \begin{cases} = 0 & \text{if} \quad (u, v) \in \mathcal{V}_F \times \mathcal{V}_F, \\ < T & \text{if} \quad (u, v) \in \mathcal{E}_C, \\ > T & \text{otherwise.} \end{cases} \tag{20}$$

The desired VBC-Box embedding $\phi$ for $G$ with $d = d_F + d_C = \mathcal{O}((\Delta_F + 2)\log(|\mathcal{V}_F|) + |\mathcal{V}_C|^2)$ dimensions now be created by concatenating $\phi_1$ and $\phi_2$, for which

$$\mathrm{E}_{\text{VBC-Box}}(u, v; \phi, (\tau, \nu)) = \mathrm{E}_{\text{VBC-Box}}(u, v; \phi_1, (\tau, \nu)) + \mathrm{E}_{\text{VBC-Box}}(u, v; \phi_2, (\tau, \nu)). \tag{21}$$

Since $\mathcal{E}_F \cap \mathcal{E}_C = \emptyset$, inspecting (19) and (20) we have that

$$\mathrm{E}_{\text{VBC-Box}}(u, v; \phi(\tau, \nu)) < \mathcal{T} \quad \Longleftrightarrow \quad (u, v) \in (\mathcal{E}_C \cap \mathcal{E}_{\neg F}) \cup (\mathcal{E}_F \cap (\mathcal{V}_F \times \mathcal{V}_F)) = \mathcal{E}.$$

$\square$

# E   Data Statistics

**Google** [16] (15,763 nodes and 171,206 edges) is a hyperlink network from pages within Google's sites. Nodes represent pages and directed edges represent hyperlink between pages. **Epinions** [19] (75,879 nodes and 508,837 edges) is a trust network from the online social network Epinions. Nodes represent users and directed edges represent trust between users. **CORA** [24] (23,166 nodes and 91,500 edges) is a citation network of academic papers. Nodes represent papers and directed edges represent the citation relationships between papers. **Twitter** [5] (465,017 nodes and 834,797 edges) is a social network. Nodes represent users and directed edges represent following relationships between users. **DREAM5** [12] is a gene regulatory networks across organisms. *In silico* network has 1,565 nodes and 4,012 edges. *E. Coli* network has 1,081 nodes and 2,066 edges. *S. cerevisiae* network has 1,994 nodes and 3,940 edges. These networks contain a relatively small number number of cycles.

# F   Link Prediction on DREAM5 Datasets

In Table 4, we compared with a recent work that embeds graphs into pseudo-Riemannian manifolds [23], along with other baselines such as Euclidean and Hyperboloid embeddings on DREAM5 datasets. Models are evaluated using Average Precision (AP). Results show that Binary Code Box significantly out-performs baseline methods on *In Silico* and *S. Cerevisiae* datasets, while showing competitive performance on the *E. Coli* graph. It can also be observed that Gumbel Box performs competitively on these graphs, which are almost acyclic.

Table 4: **Link prediction on Experiments on DREAM5 Datasets**. Following Sim et al. [23], we use median Average Precision among 5 test sets with different negative samples as evaluation metric, and we sample 4 times the negatives by randomly corrupting one of the node in each positive edges in the test set. We compare our model with other baselines using 10, 50, 100 number of parameters per vertex. Cyclicity and transitivity of each graph are shown in the table. Bold numbers perform the best, and underscored numbers perform the best in all non-box models. For more details of baselines models, please refer to [23].

| Methods | In Silico | | | E. Coli | | | S. Cerevisiae | | |
|---|---|---|---|---|---|---|---|---|---|
| Cyclicity = | 0.01 | | | 0.01 | | | 0.01 | | |
| Transitivity = | 0.25 | | | 0.40 | | | 0.17 | | |
| # parameters / vertex | 10 | 50 | 100 | 10 | 50 | 100 | 10 | 50 | 100 |
| Euclidean + FD | 39.7 | 39.8 | 34.8 | 40.2 | 44.5 | 49.0 | 40.2 | 44.5 | 49.0 |
| Hyperboloid + FD | 50.8 | 50.9 | 52.5 | 52.7 | 53.6 | 50.6 | 46.5 | 48.8 | 47.9 |
| Minkowski + TFD | 51.2 | 57.7 | 58.0 | 63.4 | 67.7 | **68.2** | 46.4 | 52.7 | 54.0 |
| Anti de-Sitter + TFD | 51.9 | 55.6 | 56.0 | 61.8 | 63.3 | 63.0 | 44.9 | 47.5 | 49.4 |
| Cylindrical Minkowski + TFD | 56.3 | 58.9 | 61.0 | 62.3 | 65.8 | 63.2 | 46.8 | 53.4 | 54.6 |
| Vector(Ours)* | 56.7 | 59.2 | 59.8 | 56.0 | 58.1 | 59.6 | 51.2 | 55.2 | 55.2 |
| G-Box | **62.3** | 66.1 | 66.6 | 65.1 | 66.5 | 68.0 | 55.0 | 58.6 | 59.5 |
| GBC-Box | 62.0 | 66.4 | **68.8** | **65.4** | **68.3** | 65.9 | 55.1 | 59.4 | **59.7** |
| VBC-Box ($d_{bin} < d$) | 58.4 | **66.5** | 66.4 | 62.6 | 67.3 | 66.1 | 52.1 | **59.6** | 58.3 |
| VBC-Box ($d_{bin} = d$) | 55.3 | 66.0 | 64.9 | 58.1 | 65.8 | 65.3 | **55.3** | 57.5 | 57.7 |

## G  Hyper-parameter Search

We follow the setting from Boratko et al. [2] for hyper-parameter search strategies in graph reconstruction experiments. Table 5 shows ranges of Bayesian hyper-parameter search for our link prediction experiments.

Table 5: Hyper-parameter range of Bayesian optimization for link prediction.

| Hyper-Parameter | Range |
|---|---|
| learning rate | 1e-5 $\sim$ 1e-2 |
| batch size | 1024 (Table 3), 64 (Table 4) |
| max epochs | 16, 32, 64, 128 |
| $\tau$ | 0.001 $\sim$ 0.1 |
| $\nu$ | 0.1 $\sim$ 10.0 |
| $w_r$ | $10^{-8} \sim 1.0$ |
| $k$ | 1 $\sim$ 10 |
| $d_{bin}$ | 0 $\sim$ d |

## H  Case Study

In this section, we visualize how binary codes work to preserve transitivity and cyclicity together.

As shown in Figure 4, our analysis is over two synthetic graphs. For fair comparison, we embed both graphs into 3 dimensional G-BOX and 2 dimensional VBC-BOX. The graph on the top is formed by a 7-node directed chain (vertex 0 to vertex 6) with full transitive closures and one additional node connecting the chain into a cycle. It shows that Gumbel Box can model transitive closures well, but cannot model the cycle while Binary Code Box can handle both. The latter models transitive closures by sharing full box space from node 0 to node 5, and then models the cycle by selecting sub-dimensions in node 6 and 7. The graph on the bottom is formed by a chain of triangle cycles. It shows that Gumbel Box cannot handle cycles, and generates an acyclic graph. In contrast, binary code boxes can handle this graph nicely with much lower errors by alternately switching among sub-spaces within each triangle cycles.

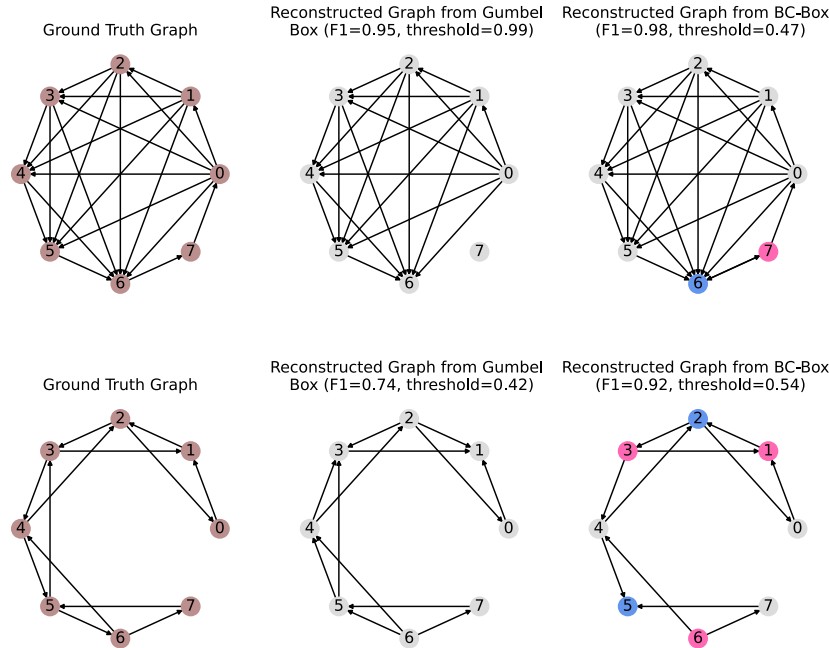

Figure 4: Two directed graphs and reconstructed graphs by G-Box (3D) and VBC-Box (2D). The upper graph is a directed cycle with almost all transitive closures. The bottom graph is a chain of directed triangle cycles. Blue colored vertices have binary codes of [1,0], pink colored vertices have binary codes of [0,1], and grey colored vertices have binary codes of [1, 1].

In Fig. 5, we visualize how GBC-Box handles cycles and transitive edges. We compare GBC-Box with 2 dimensional G-Box. Fig. 5a shows that when representing a DAG, GBC-Box learns to utilize all dimensions in both binary code vectors and leverages box containment to model edge directions. Fig. 5b shows that given a pure cycle, GBC-Box learns "skinny" boxes to model $0 \rightarrow 1, 2 \rightarrow 3$ using the vertical axis, and the rest of the edges in the horizontal axis. Fig. 5c shows a more complicated graph and our model also learns to split the graph into $0 \rightarrow 1, 1 \rightarrow 2, 0 \rightarrow 2$ in the horizontal axis, and $2 \rightarrow 3, 3 \rightarrow 0, 2 \rightarrow 0$ in the vertical axis. In comparison, the original G-Box struggles with cycles and cannot reconstruct ground truth graphs from 5b and 5c [7].

---

[7]The 0th and 2nd boxes cover almost same regions in the second row of Figure 5c

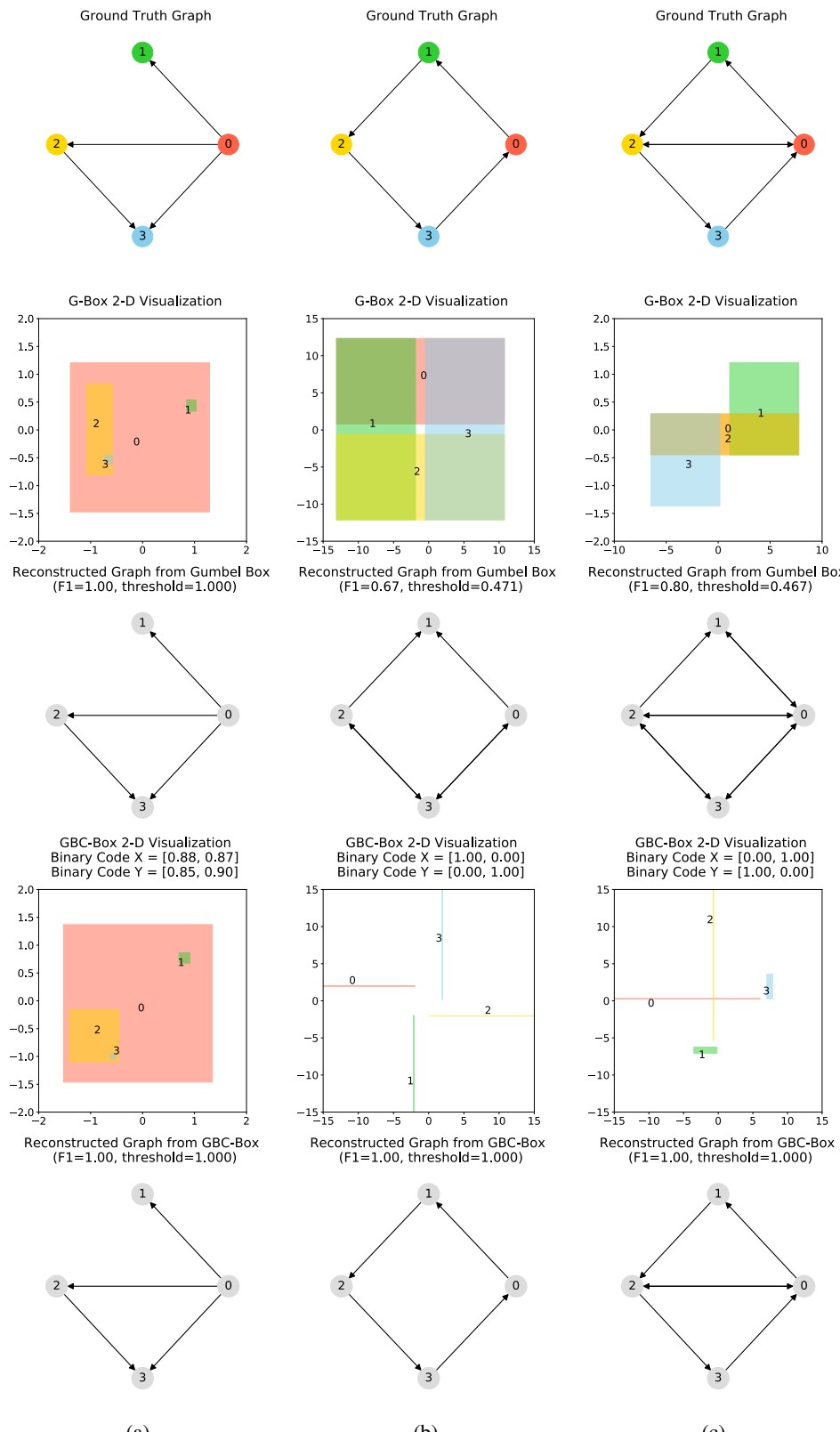

Figure 5: Visualization of graph reconstruction using 2 dimensional G-Box and GBC-Box. In figure 5a, the ground truth graph is a DAG (an out-tree with transitive closure). In figure 5b, we have a pure cycle. In figure 5c, we have three cycles nested together, forming two 2-hop transitive closures. We visualized 2-D boxes trained via gradient descent and binary cross entropy loss with $\tau = 0.001$ and $\nu = 0.5$, and learning rate 0.01.