# OpenReview forum: "Modeling Transitivity and Cyclicity in Directed Graphs via Binary Code Box Embeddings"
_NeurIPS.cc/2022/Conference — NeurIPS 2022 Accept_

### Official Review · Reviewer_f6GD · 2022-07-05

**Rating:** 5
**Confidence:** 3
**Soundness:** 2 fair
**Presentation:** 3 good
**Contribution:** 3 good

**Summary:**

This paper proposed an extension of box embedding on dealing with directed graphs with cycles. The idea is called binary code box embedding, which learns to project the box embeddings into a subspace of the box space. The authors prove the theoretical expressivity of the proposed model. Empirical results on two tasks further show that the model provides useful inductive biases for preserving transitivity and cyclicity.

**Questions:**

The author in [1] proved that box embedding has some theoretical limitations on embedding some directed graphs (e.g., a counterexample is given in section 7.3 of [1]). How does the proposed model encode such a counterexample?

[1] Vilnis, L., 2021. Geometric Representation Learning.

**Limitations:**

The authors discussed the limitations of the proposed Binary Code Box Embeddings and proposed a regularization term to alleviate but not fully address such problem.

**Strengths And Weaknesses:**

Pros

1. An interesting extension of box embeddings for modeling cyclicity
2. Strong empirical results

Cons

1. Theoretical results in Sec 3.1 are taken from [2]. The author should cite it right after the title like “Theorem 1 [2]” to avoid misunderstanding.
2. Theoretical results in Sec 3.2 are a bit trivial. It seems very clear that the box embedding cannot model any cycles in any dimension.
3. The motivation for using binary code embedding is not very clear. Could you give an example of how the projection models cycles? For instance, how binary code box embedding can embed a pure cycle with 4 nodes with minimal dimension? that would be a nice picture that gives readers an intuition of the idea.

---

> ### Author Response · Authors · 2022-08-02
> **Thank you for your insightful comments! We have answered your questions in the comment below.**
>
> > Theoretical results in Sec 3.1 are taken from [2]. The author should cite it right after the title like “Theorem 1 [2]” to avoid misunderstanding.
>
> Theorem 1 is significantly different from that presented in [2]. In [2], the authors actually only proved that hard box (i.e. G-Box with temperature = 0) can model any DAG. Theorem 1 in our work proves that G-Box can represent any DAG that hard box can represent. The proof is non-trivial, and necessary not only to prove the representational capacity of BC-Box (by extending from “hard” to “soft” temperatures), but also proves additional representational capacity of the G-Box model. Please refer to Appendix B for more details.
>
> > Theoretical results in Sec 3.2 are a bit trivial. It seems very clear that the box embedding cannot model any cycles in any dimension.
>
> The claim that box embeddings cannot model cycles in any dimension is a bit more nuanced than it may first appear. For example, the requirement of “chordless” is crucial. In fact, in our Proposition 1, we showed that G-Box can represent a graph which is the union of a chordless cycle and one reverse edge.
>
> > The motivation for using binary code embedding is not very clear. Could you give an example of how the projection models cycles? For instance, how binary code box embedding can embed a pure cycle with 4 nodes with minimal dimension? that would be a nice picture that gives readers an intuition of the idea.
>
> In this link (https://imgur.com/a/Zm9hP09), we visualize how GBC-Box handles cycles and transitive edges. We compare GBC-Box with 2-dimensional G-Box. Fig. 4a shows that when representing a DAG, GBC-Box learns to utilize all dimensions in both binary code vectors and leverages box containment to model edge directions. Fig. 4b shows that given a pure cycle, GBC-Box learns “skinny” boxes to model 0 → 1, 2 → 3 using the vertical axis, and the rest of the edges in the horizontal axis. Fig. 4c shows a more complicated graph. Similarly, our model learns to split the graph into 0 → 1, 1 → 2, 0 → 2 in the horizontal axis, and 2 → 3, 3 → 0, 2 → 0 in the vertical axis. In comparison, the original G-Box struggles with cycles and cannot reconstruct ground truth graphs from 4b and 4c (The 0th and 2nd boxes cover almost same regions in the second row of Figure 4c). We also updated this to the new appendix pdf in the zip file.
>
> > The author in [1] proved that box embedding has some theoretical limitations on embedding some directed graphs (e.g., a counterexample is given in section 7.3 of [1]). How does the proposed model encode such a counterexample?
>
> The limitation mentioned in [1] is with regard to probability modeling and does not directly apply to graph modeling. As we prove in this work, there is *no* theoretical limitation to modeling directed graphs. That being said, if pressed, we could ask a similar question:
>
> Can our model represent:
>
>  $\text{Vol}(\text{Box}(A)\cap\text{Box}(B)) = \text{Vol}(\text{Box}(A)\cap\text{Box}(C)) =\text{Vol}(\text{Box}(B)\cap\text{Box}(C)) = \epsilon$ and $\text{Vol}(\text{Box}(A)\cap\text{Box}(B)\cap\text{Box}(C))= 0$, where $\epsilon > 0$ ?
>
> Assume that we create two global binary code vectors and generate two independent spaces using $\text{Box}_1()$ and $\text{Box}_2()$, and the overall score of box intersection volume equals to the maximum score from two spaces.
>
> Then, we can construct $\text{Box}_1$ and $\text{Box}_2$ such that
>
> $\text{Vol}(\text{Box}_1(A)\cap\text{Box}_1(B)) = \epsilon$, $\text{Vol}(\text{Box}_1(A)\cap\text{Box}_1(C)) = \epsilon$ and $\text{Vol}(\text{Box}_1(B)\cap\text{Box}_1(C)) = 0$;
>
> $\text{Vol}(\text{Box}_2(A)\cap\text{Box}_2(B)) = 0$, $\text{Vol}(\text{Box}_2(A)\cap\text{Box}_2(C)) = \epsilon$ and $\text{Vol}(\text{Box}_2(B)\cap\text{Box}_2(C)) = \epsilon$
>
> Then, we can derive that
>
> $\text{Vol}(\text{Box}(A)\cap\text{Box}(B)) = \max(\text{Vol}(\text{Box}_1(A)\cap\text{Box}_1(B)), \text{Vol}(\text{Box}_2(A)\cap\text{Box}_2(B))) = \epsilon$,
>
> $\text{Vol}(\text{Box}(A)\cap\text{Box}(C)) = \max(\text{Vol}(\text{Box}_1(A)\cap\text{Box}_1(C)), \text{Vol}(\text{Box}_2(A)\cap\text{Box}_2(C))) = \epsilon$,
>
> $\text{Vol}(\text{Box}(B)\cap\text{Box}(C)) = \max(\text{Vol}(\text{Box}_1(B)\cap\text{Box}_1(C)), \text{Vol}(\text{Box}_2(B)\cap\text{Box}_2(C))) = \epsilon$
>
> And
>
> $\text{Vol}(\text{Box}(A)\cap\text{Box}(B)\cap\text{Box}(C)) = $
>
> &nbsp; &nbsp; &nbsp; &nbsp; &nbsp; &nbsp; &nbsp; &nbsp; &nbsp; &nbsp; &nbsp; &nbsp; &nbsp;
> $\max(\text{Vol}(\text{Box}_1(A)\cap\text{Box}_1(B)\cap\text{Box}_1(C)), $
>
> &nbsp; &nbsp; &nbsp; &nbsp; &nbsp; &nbsp; &nbsp; &nbsp; &nbsp; &nbsp; &nbsp; &nbsp; &nbsp;
> $\text{Vol}(\text{Box}_2(A)\cap\text{Box}_2(B)\cap\text{Box}_2(C))) = 0$
>
>
> Therefore this analogous scenario *can* be handled via binary codes. However, we agree that exploring higher-order representations and their potential limitations is an interesting line of future work for *BC-Box models.
>
> *[1] Vilnis, L., 2021. Geometric Representation Learning.*

---

> > ### Comment · Reviewer_f6GD · 2022-08-04
> > **Further questions**
> >
> > Thank you for the detailed feedback.  I have some further questions:
> >
> > **Justification of Novelty of Theorem 1.** Why proving that G-Box (soft box) can represent any DAG is more challenging than proving that of the hard box? Given that hard Box (i.e., G-Box with with temperature = 0) can represent any DAG, I would image that soft box can also represent any DAG as hard box is just a special case of the G-Box. I didn't see any significant difference by extending from “hard” to “soft” temperatures. It seems more like a corollary instead of a theorem. The author must not exaggerate their nolvety (it is not wrong to use other's theorem but should cite properly).
> >
> > **The limitation mentioned in [1] is with regard to probability modeling and does not directly apply to graph modeling.**
> > This claim is a bit untrue, as we could easily construct a similar counter example in directed graph modeling.
> > For example, given a DAG {(4,1), (4,2),(5,2),(5,3), (6,1), (6,3)} where (i,j) means a directed edge.
> > The box model will enforce node 4,5,6 to be inside the intersection of box (1,2), (2,3), and (3,1) respectively.  However, it will also enforce the existence of one of the edge (6,2), (5,1), (4,3) which we do not want. Is box (G-Box) able to represent such graph?

---

> > > ### Author Response · Authors · 2022-08-07
> > > **Thank you for your further comments. We have answered both of your questions below**
> > >
> > > > **Justification of Novelty of Theorem 1**
> > >
> > > We respectfully disagree - the theoretical results in section 3.1 are *not* taken from [2], nor are they immediate corollaries of the results in [2]. We summarize and properly cite the results from [2] in lines 105-107. We agree that one could obtain the following immediate corollary from the results in [2] by leveraging the equivalence of their model with our $\text{G-Box}$ model (as proved in Appendix A):
> > >
> > > **Corollary 1:** Let $G$ be any DAG. Then for $\tau = \nu = 0$, there exists a mapping $\phi: \mathcal V \to \mathcal R^{2d}$ and a threshold $T > 0$ such that
> > > $\text{E}_\text{G-Box}(u,v; \phi, (\tau, \nu)) < T$ if and only if $(u,v) \in \mathcal E$,
> > > where $d = \mathcal O((\Delta + 2) \log |\mathcal V|)$, and $\Delta$ is the maximum degree in $G$.
> > >
> > > Our statement, however, removes the requirement that $\tau = \nu = 0$, essentially turning a **“there exists”** statement into a **“for all”** statement:
> > >
> > > **Corollary 2:** Let $G$ be any DAG. **Given any** $\tau = \nu  \in \mathbb R_{\ge 0}$, there exists a mapping $\phi: \mathcal V \to \mathbb R^{2d}$ and a threshold $T > 0$ such that
> > > $\text{E}_\text{G-Box}(u,v; \phi, (\tau, \nu)) < T$ if and only if $(u,v) \in \mathcal E$,
> > > where $d = \mathcal{O}((\Delta + 2) \log |\mathcal V|)$, and $\Delta$ is the maximum degree in $G$.
> > >
> > > Changing an existential quantifier to a universal one is rarely trivial, and in this case it requires 2 pages of detailed arguments (see Appendix B). The proof itself also provides several useful insights to the $\text{G-Box}$ model (eg. Proposition 3, which establishes that the representational capacity of $\text{G-Box}$ depends only on the ratio of $\tau$ and $\nu$, and not their absolute value), and this stronger result is necessary in order to establish the representative capacity of $\text{GBC-Box}$ (Theorem 5).
> > >
> > > > **The limitation mentioned in [1]**
> > >
> > > The example you provided is, indeed, a directed acyclic graph, and so (as you correctly observed previously) we know it *can* be represented using $\text{G-Box}$. Here is a construction based on the proof in [2] (https://imgur.com/a/PoytBgB).
> > >
> > > (Note: in order to represent the scenario where "The box model will enforce node 4,5,6 to be inside the intersection of box (1,2), (2,3), and (3,1) respectively", we should actually reverse edges' direction ( {(4,1), (4,2),(5,2),(5,3), (6,1), (6,3)} -->  {(1,4), (2,4),(2,5),(3,5), (1,6), (3,6)} ), according to the energy definition in [2], eq 14. Of course, we can model {(4,1), (4,2),(5,2),(5,3), (6,1), (6,3)} similarly, as these two graphs are isomorphic.)
> > >
> > > More generally, the limitation mentioned in [1] cannot possibly impose a limitation for either $\text{BC-Box}$ model when representing directed graphs, as Theorem 5 and 6 prove that any directed graph can be represented.
> > >
> > >
> > > [1] Vilnis, L., 2021. Geometric Representation Learning.
> > >
> > > [2] Boratko, M. et al., NeurIPS 2021. Capacity and Bias of Learned Geometric Embeddings for Directed Graphs.

---

### Official Review · Reviewer_3KMQ · 2022-07-12

**Rating:** 5
**Confidence:** 4
**Soundness:** 3 good
**Presentation:** 4 excellent
**Contribution:** 2 fair

**Summary:**

The authors consider the problem of modeling directed graphs for semi-supervised link prediction. They propose a new model based on an extension of box embeddings which enables greater representational capacity, particularly in the presence of cycles. This extension may be viewed as a decomposition of a graph into a finite union of graphs, each of which modeled using (modified) box embeddings. The authors show that, with sufficiently high-dimensional embeddings, this model may represent an arbitrary directed acyclic graph. The authors demonstrate the effectiveness of their model through some experiments on synthetic graph models, and some applications on real data.

**Questions:**

It is unclear to me how the baseline vector model differs from existing pairwise latent embedding models such as node2vec. Indeed, it seems to me that the edge likelihood is exactly the same in either case (modulo directedness, which can be dealt with in such representations by separating the source and target embedding) - the crucial difference being the way edges are sampled and the effective induced weighting through the sampling procedure. Would the authors be able to describe in more detail what the exact loss for the vector model is?

In the paragraph preceding table 3, the authors mention that they evaluate t-Box, but this does not seem to appear in the table. Could the authors share these numbers and add them to the table?

I was confused by the accounting of the number of parameters in the binarycode models in the provided code in the supplementary. My reading of the current code (`model.py:375-378`) indicates that the total number of parameters is $3 dim$. However, in `main.py:33` the computation of `dim` as a function of `num_parameters` and `shared_dim` gives, e.g. when `shared_dim = num_parameters / 2` that `dim = num_parameters / 2`, leading to a total of 50% more parameters than claimed. Running the code and inspecting the generated embedding tensors seems to support that claim. Would the authors be able to comment on how the parameters are counted for the binary box models?


**Limitations:**

I did not find any particular limitation except as described in the weaknesses. I do not believe this work has any particular societal impact.

**Strengths And Weaknesses:**

Overall, I found the proposed method interesting and with good empirical performance. The authors propose an elegant solution to address issues with existing geometric embedding models. However, I also found this work to be fairly incremental (especially with respect to [2]), both from a theoretical and an empirical perspective, and had some doubts on the general motivation of the paper. Finally, I have some concerns in the empirical evaluations in the paper as it currently stands (see questions). On the balance, I do not feel that this work is currently a substantial enough contribution to warrant publication, although I am willing to revisit this judgment should my questions be addressed.

The modification proposed by the authors is elegant and addresses a potential issue in the representational power of box embeddings for graphs, namely the ability to model directed cycles. However, it is not clear to me how this relates to the claims around transitivity. Indeed, it seems to me that in the setting considered by the authors, transitivity is viewed not as a symbolic requirement, but rather a feature of the data (“soft transitivity”). In that setting, the argument given by the authors on ll. 34-35, that there is no encouragement for transitivity to hold seems suspicious: given that such edges are present in the data, it is unclear why a purely data-based approach is not able to recover them.

Although the proposal of the authors is well-motivated theoretically, I found the empirical results to be unconvincing. Indeed, in table 3, the authors show that the G-box model achieves the same performance for 3 of the 4 benchmarks, and is only slightly behind on the 4th one. In practice, it thus seems that the cycle representation is not an issue, and the model is able to disambiguate cycles based on additional context. Especially given the additional flexibility in the *BC-box models, would the authors be able to provide additional empirical evidence that the difference, when it exists, is due to the inability to model cycles?

Typos:
L. 36 $o_k$ -> $t_k$
L. 90 it’s -> its

---

> ### Author Response · Authors · 2022-08-02
> **Thank you for your insightful comments!  We have answered your questions in the comment below.**
>
> >given that such edges are present in the data, it is unclear why a purely data-based approach is not able to recover them.
>
> Yes, in fact, with sufficient dimensionality, a source-target vector inner product model can recover any graph, including one with transitive edges, as observed in both [2] and Table 2 in our work, where the vector-based model captures the transitive closure edges perfectly, given enough dimensions. However, (1) this may require significantly more dimensions than box embeddings (c.f. [2] and our Table 2); (2) such a vector model (with separate source-target vectors for each node) would have little to no generalization ability. As shown in Table 2, under the “w. transitive closures ” setting, and 12 parameters (6 dimensions for vector and 6 dimensions for G-Box),  G-Box and GBC-Box models can perfectly recover all edges, while vector-based model *cannot*.
>
> > Especially given the additional flexibility in the *BC-box models, would the authors be able to provide additional empirical evidence that the difference, when it exists, is due to the inability to model cycles?
>
> We provide evidence of this in two settings:
>
> **Reconstruction:** We verified that *BC-Box models are better at capturing cycles in our reconstruction experiments, where the entire graph is known. In particular, in Table 1 we see that VBC-Box > GBC-Box > G-Box when reconstructing a given graph, and in Figure 1: we showed that our models out-perform G-Box more when there are more cycles. This is a more suitable setting to address the question of the model’s ability to capture cycles, as we have the true ground-truth graph available.
>
> **Generalization:** We also provide evidence (Table 3) that the model is better at capturing cycles in the generalization setting - for example, the performance increase on Epinions (cyclicity = 0.88) is much greater than that on CORA (cyclicity = 0.23), and this pattern also holds true (to a lesser degree) for Google (cyclicity = 0.96) vs Twitter (cyclicity = 0.01). In principle, however, it is more difficult to attribute the increase in performance for the generalization experiments to the ability of the model to represent cycles, because it is unclear the extent to which modeling cycles is relevant for predicting the specific edges in the test set.
>
> > Would the authors be able to describe in more detail what the exact loss for the vector model is?
>
> It is true that our vector baseline is essentially the same as node2vec. Specifically, the energy is $E(u,v) = - \log (\sigma(\phi_\text{out}(u), \phi_\text{in}(v)))$, and we use binary cross-entropy loss. The main reason for our reimplementation of a vector baseline was to ensure that only the energy function was different, with equal hyper-parameter tuning and negative sampling.
>
> For all models we trained, the **negative samples** were chosen from **non-training edges**, which differs from the original node2vec implementation which samples nodes from the full vocabulary. This filtering can result in a large improvement, as shown in TransE work (*Translating Embeddings for Modeling Multi-relational Data*, NIPS 2013) where filtering (denoted as “filt” in Table 3 of TransE paper) improved the performance of Hits@10 from **34.9 to 47.1**.
>
> We also wish to clarify that results in the first 6 rows from Table 3 are from [30], therefore it is not clear whether they have certain modifications or the extent to which they were tuned. Our intent was to provide a rigorous apples-to-apples comparison, and for that reason, we trained our own version of a vector-based model.
>
> *[30] Shijie Zhu et al. Adversarial directed graph embedding. AAAI 2021*
>
> > In the paragraph preceding table 3, the authors mention that they evaluate t-Box, but this does not seem to appear in the table. Could the authors share these numbers and add them to the table?
>
> We added the t-Box performance in the table below:
> | Model |  Google | Google | Epinions | Epinions | CORA | CORA |  Twitter | Twitter |
> |---------|-----------|-----|--------|-------|-------|------|------|----------|
> |G-Box| 99.2 | 98.2 | 95.1 | 90.0 | 93.9 | 89.6 | 99.8 | 86.3  |
> |**t-Box** |               97.1 | 95.8 | 96.4 | 89.6 | 87.3 | 79.6 | 99.8 | 84.4 |
>
> It is somewhat surprising that t-Box is lower than G-Box, however, we posit that perhaps the flexibility of t-Box is allowing it to overfit.
>
>
> > Would the authors be able to comment on how the parameters are counted for the binary box models?
>
> When `shared_dim = num_parameters / 2`, `self.codes` becomes an empty matrix, and since `self.ones` is a constant matrix, it is not taken into account the number of parameters (Was this the part that misled you?). Therefore, the model is equivalent to Gumbel Box in dimension `num_parameters / 2`.
>
> All our experiments compare baselines with an equal **number of parameters per node**, rather than dimensions, in order to provide an apple-to-apple comparison. Thank you for double-checking our code though! We appreciated it!

---

> > ### Comment · Reviewer_3KMQ · 2022-08-08
> > **Thanks for clarifications**
> >
> > Thanks for including the additional information and clarifying my misunderstanding of your code!

---

> > > ### Author Response · Authors · 2022-08-08
> > > **Any Further Questions?**
> > >
> > > You're quite welcome. We hope we addressed your questions sufficiently! If so, we would greatly appreciate reconsideration of the rating. We would, of course, be happy to answer any further questions and are open to additional suggestions we can make to improve the submission further. Thank you!

---

### Official Review · Reviewer_99se · 2022-07-17

**Rating:** 6
**Confidence:** 3
**Soundness:** 3 good
**Presentation:** 3 good
**Contribution:** 3 good

**Summary:**

Introduces the concept of binary code box embeddings and theoretically shows that the proposed models have the expressive power to model arbitrary directed cyclic graphs. Empirical result suggests that the proposed models perform better on 4 link prediction datasets compared to vector based baselines.

**Questions:**

- The paper does a good job analysing the expressive power of the model. However, as the author(s) mention, whether or not the model can be trained via gradient descent is not yet clear. The paper does not currently have details on the optimization challenges/ease of the proposed models vs the other baselines. It is often possible that the expressive power is sufficient, but the optimization difficulty is high.

- While the intuition of binary code box embedding bases off box embeddings, and has a nice theoretical picture, it is hard to intuitively see how much of the geometry structures are still learned by the models, since binary codes break transitivity. It would be interesting to see a visual comparison of the learned boxes in 2D with and without binary codes.



**Limitations:**

How are the binary code represented during training/inference? Are they represented as a probability since the model is trained using gradient descent, or are they pushed to 0 and 1s?

**Strengths And Weaknesses:**

- Binary code box embeddings is a novel concept with nice theoretical intuitions. It relieves some of the constraints of box embeddings at the expense of weakening transitivity.


- Showed both theoretically and empirically the advantage of the binary code models being able to model directed cyclic graphs.


- Provided nice intuition for proposed concepts using interval graphs, including box embeddings can be thought of representing a graph as an intersection over some learned subgraphs. Binary codes can be thought of redefining “intersection”, such that two non-overlapping boxes can be seen as overlapping by selecting which dimensions to ignore using the binary codes.

---

> ### Author Response · Authors · 2022-08-02
> **Thank you for your insightful comments!  We have answered your questions in the comment below.**
>
> > The paper does a good job analysing the expressive power of the model. However, as the author(s) mention, whether or not the model can be trained via gradient descent is not yet clear. The paper does not currently have details on the optimization challenges/ease of the proposed models vs the other baselines. It is often possible that the expressive power is sufficient, but the optimization difficulty is high.
>
> Our experiments indicate that the model can be easily trained via gradient descent. It is difficult to make any theoretical claims regarding ease of training, of course, but after having trained the model on many synthetic and real-world graphs (see section 5), we do not observe any additional difficulty when training our model via gradient descent as compared to (say) G-Box or t-Box models.
>
> >While the intuition of binary code box embedding bases off box embeddings, and has a nice theoretical picture, it is hard to intuitively see how much of the geometry structures are still learned by the models, since binary codes break transitivity. It would be interesting to see a visual comparison of the learned boxes in 2D with and without binary codes.
>
> We agree, that it is difficult to envision how the geometry of the binary code boxes may capture various graphs, and unfortunately (both with traditional box embeddings and binary code boxes), an intuition gained in 2 dimensions often is misleading for higher dimensions. That being said, our understanding and intuition behind the model are to approach it from the level of operations on interval graphs, as described in section 4.1, which is also historically how box representations for graphs were motivated (a la boxicity).
>
> In this link ([https://imgur.com/a/Zm9hP09](https://imgur.com/a/Zm9hP09)), we visualize how GBC-BOX handles cycles and transitive edges. We compare GBC-BOX with 2-dimensional G-BOX. Fig. 4a shows that when representing a DAG, GBC-BOX learns to utilize all dimensions in both binary code vectors and leverages box containment to model edge directions. Fig. 4b shows that given a pure cycle, GBC-BOX learns “skinny” boxes to model 0 → 1, 2 → 3 using the vertical axis, and the rest of the edges in the horizontal axis. Fig. 4c shows a more complicated graph. Similarly, our model learns to split the graph into 0 → 1, 1 → 2, 0 → 2 in the horizontal axis, and 2 → 3, 3 → 0, 2 → 0 in the vertical axis. In comparison, the original G-BOX struggles with cycles and cannot reconstruct ground truth graphs from 4b and 4c (The 0th and 2nd boxes cover almost same regions in the second row of Figure 4c). We also updated this to the new appendix pdf in the zip file.
>
> > How are the binary code represented during training/inference? Are they represented as a probability since the model is trained using gradient descent, or are they pushed to 0 and 1s?
>
> We represent binary codes using logits (i.e. training free parameters which are put through a sigmoid function). After training, binary codes are often either close to 1 or 0. We use them directly for inference without ‘’binarization”.

---

### Meta-Review · Area_Chair_HMGf · 2022-08-27

**Recommendation:** Accept
**Confidence:** Less certain

**Metareview:**

This paper extends box embeddings to allow for them to represent directed graphs. In particular, the proposed binary box embeddings can represent cycles in graphs, which was not previously possible. The reviewers appreciated the introduction of binary box embeddings and felt the contribution was novel and elegant. During the discussion, the reviewers felt that the rebuttal generally answered their questions and felt that the contribution would be of interest to the NeurIPS community.

A number of clarity issues brought up in the initial reviews should be addressed for the final version of the work, for instance, the motivation of the binary code embeddings and the discussions with 3KMQ on transitivity holding. Please revise the paper to address the remaining comments from the reviewers and carefully incorporate the additional results presented during the author response discussion phase.

**Award:**

No

---

### Decision · Program_Chairs · 2022-09-14

Accept